# Phosphoinositide-mediated oligomerization of a defensin induces cell lysis

Ivan KH Poon[†], Amy A Baxter[†], Fung T Lay[†], Grant D Mills, Christopher G Adda, Jennifer AE Payne, Thanh Kha Phan, Gemma F Ryan, Julie A White, Prem K Veneer, Nicole L van der Weerden, Marilyn A Anderson, Marc Kvansakul*, Mark D Hulett*

Department of Biochemistry, La Trobe Institute for Molecular Science, La Trobe University, Melbourne, Australia

**Abstract** Cationic antimicrobial peptides (CAPs) such as defensins are ubiquitously found innate immune molecules that often exhibit broad activity against microbial pathogens and mammalian tumor cells. Many CAPs act at the plasma membrane of cells leading to membrane destabilization and permeabilization. In this study, we describe a novel cell lysis mechanism for fungal and tumor cells by the plant defensin NaD1 that acts via direct binding to the plasma membrane phospholipid phosphatidylinositol 4,5-bisphosphate ($PIP_2$). We determined the crystal structure of a NaD1:$PIP_2$ complex, revealing a striking oligomeric arrangement comprising seven dimers of NaD1 that cooperatively bind the anionic headgroups of 14 $PIP_2$ molecules through a unique 'cationic grip' configuration. Site-directed mutagenesis of NaD1 confirms that $PIP_2$-mediated oligomerization is important for fungal and tumor cell permeabilization. These observations identify an innate recognition system by NaD1 for direct binding of $PIP_2$ that permeabilizes cells via a novel membrane disrupting mechanism.

*For correspondence: m.hulett@latrobe.edu.au (MDH); m.kvansakul@latrobe.edu.au (MK)

[†]These authors contributed equally to this work

## Introduction

Host defense peptides, which include cationic antimicrobial peptides (CAPs), are a group of innate immune molecules produced by essentially all plant and animal species that act as a first line of defense against microbial invasion. In common with most innate immunity peptides, they are relatively small (typically <100 amino acid residues), are predominantly cationic, and typically harbor a substantial number of hydrophobic amino acids (*Hancock and Lehrer, 1998*; *Brogden, 2005*; *Lai and Gallo, 2009*). Although originally identified due to their potent activity against microbial pathogens, several CAPs also exhibit cytolytic activity against a range of mammalian tumor cells (*Lichtenstein et al., 1986*; *Cruciani et al., 1991*; *Hancock and Sahl, 2006*; *Schweizer, 2009*).

The defensins are a family of CAPs that are ubiquitously expressed in plants, animals, insects, and fungi that play an important role in innate immune defense against microbial threats (*Brogden, 2005*; *Lay and Anderson, 2005*; *Hancock and Sahl, 2006*; *Lai and Gallo, 2009*). The plant defensins belong to a large family of molecules that are highly variable in sequence but have a conserved structure. The sequence variability leads to several biological functions including antimicrobial activity, regulation of plant development, and pollen tube guidance (*Carvalho and Gomes, 2009*; *De Coninck et al., 2013*). Even those plant defensins that have been ascribed antifungal activity have large differences in sequence and are likely to act by different mechanisms (*van der Weerden and Anderson, 2013*). The plant defensins are small (~5 kDa, 45–54 amino acids), basic, cysteine-rich proteins that display a family-defining disulfide bond array (in a $C_I$–$C_{VIII}$, $C_{II}$–$C_V$, $C_{III}$–$C_{VI}$, and $C_{IV}$–$C_{VII}$ configuration) known as the

**eLife digest** It is often said that attack is the best form of defense; and the immune systems of plants and animals will often target the cell membranes of microbes and other pathogens in order to defend themselves. Disrupting the cell membrane causes essential contents to leak from the cell, and eventually, the cell will burst and die.

Most plants and animals produce small proteins called defensins that kill microbes by attacking their cell membranes. These defensins are thought to either destabilize the cell membrane by coating its outer surface or to insert themselves into the membrane to form open pores that allow vital biomolecules to leak out of the cell. However, the exact mechanism by which defensins attack microbial membranes is not understood.

In this study, Poon, Baxter, Lay et al. show that a defensin called NaD1—which was isolated from the ornamental tobacco *Nicotiana alata*—binds to a molecule from the cell membrane called phosphatidylinositol 4,5-bisphosphate, or $PIP_2$ for short. By working out the three-dimensional structure of this complex, Poon, Baxter, Lay et al. show that it contains 14 $PIP_2$ molecules and 14 NaD1 molecules in an arch-shaped structure and suggest that sequestering large numbers of $PIP_2$ molecules in this way destabilizes the cell membrane of the microbe.

These findings raise a number of questions: are there other small proteins that can destabilize cell membranes in a similar manner to defensins? Do the immune systems of other organisms also recognize molecules from microbial cell membranes to trigger this kind of counterattack? Furthermore, since defensins can also kill tumor cells, a better understanding of how they work might also lead to new treatments for cancer and other diseases in humans.

cysteine-stabilized αβ (CSαβ) motif. This motif consists of a triple-stranded antiparallel β-sheet, which is cross-braced via three disulfide bonds at the core of the molecule to an α-helix (in a βαββ arrangement). The fourth conserved disulfide bond further rigidifies the protein by linking together the N- and C-terminal regions of the molecule, effectively generating a highly stable pseudocyclic molecule (*Janssen et al., 2003*; *Lay et al., 2003b*, *2012*; *Lay and Anderson, 2005*). This CSαβ fold is also conserved in defensins found in other organisms, including insects and fungi (*Lay and Anderson, 2005*).

NaD1, a plant defensin isolated from the flowers of the ornamental tobacco (*Nicotiana alata*), exhibits potent antifungal activity against pathogenic fungi, including *Fusarium oxysporum*, *Botrytis cinerea*, *Aspergillus niger*, *Cryptococcus species*, as well as the yeasts *Saccharomyces cerevisiae* and *Candida albicans* (*Lay et al., 2003a*, *2003b*, *2012*; *van der Weerden et al., 2008*, *2010*; *Hayes et al., 2013*). NaD1 inhibits fungal growth in a three-stage process that involves specific interaction with the cell wall and entry into the cytoplasm before cell death (*van der Weerden et al., 2008*, *2010*). Interaction with NaD1 also leads to hyper-production of reactive oxygen species, inducing oxidative damage that contributes to its fungicidal activity on *Candida albicans* (*Hayes et al., 2013*).

Many CAPs have been postulated to act at the level of the plasma membrane of target cells. Suggested mechanisms of action for membrane permeabilization are based on the (1) carpet, (2) barrel-stave, and (3) toroidal-pore models (reviewed in *Brogden, 2005*). In the carpet model, the CAPs act like classic detergents, accumulating and forming a carpet layer on the membrane outer surface, leading to local disintegration (including membrane micellization or fragmentation) upon reaching a critical concentration. Other CAPs are suggested to aggregate on the membrane surface before inserting into the bilayer forming a 'barrel-stave' pore where the hydrophobic peptide regions align with the lipid core and the hydrophilic peptide regions form the interior of the pore. Alternatively, in the toroidal pore model, the CAPs induce the lipid monolayers to bend continuously through the pore, with the polar peptide faces associating with the polar lipid head groups (*Brogden, 2005*).

Although these models have been useful for describing potential mechanisms underlying the antimicrobial activity of various CAPs, it is not clear how well they represent the actual configuration of CAPs at the membrane. Furthermore, the oligomeric state of CAPs required for their activity based on the postulated models remains unknown. Indeed, it has long been hypothesized that the molecules could form proteinaceous pores and function through insertion into membranes (*Brogden, 2005*). However, to date, the structural basis of CAP activity at the target membrane has not been defined. In

addition to the uncertainty about the configuration of CAPs at the membrane, the role of ligands in modulating the recognition of target surfaces by CAPs remains unclear.

One class of ligands that has been linked to plant defensin antifungal activity are sphingolipids (*Wilmes et al., 2011*), a key component of fungal cell walls and membranes. Plant defensins that bind sphingolipids include RsAFP2 from radish (binds glucosylceramide, GlcCer) (*Thomma et al., 2003*; *Thevissen et al., 2004*), DmAMP1 from dahlia (binds mannose-(inositol-phosphate)$_2$-ceramide, M(IP)$_2$C) (*Thevissen et al., 2000, 2003*), as well as the pea defensin Psd1 (*Goncalves et al., 2012*) and sugarcane defensin Sd5 (*de Paula et al., 2011*) that both bind membranes enriched for specific gly-cosphingolipids. MsDef1, a defensin from *Medicago sativa*, has also been implicated in binding sphingolipids, as a mutant of the fungus *Fusarium graminearum* that is depleted in glucosylceramide, is highly resistant to MsDef1 (*Ramamoorthy et al., 2007*).

In this report, we have identified the cellular phospholipid phosphatidylinositol 4,5-bisphosphate (PIP$_2$) as a key ligand that is recognized during membrane permeabilization of fungal and mammalian plasma membranes. Using X-ray crystallography, we have defined the molecular interaction of NaD1 with PIP$_2$ and demonstrate that NaD1 forms oligomeric complexes with PIP$_2$. Structure-guided muta-genesis revealed a critical arginine residue (R40) that is pivotal for NaD1:PIP$_2$ oligomer formation and that oligomerization is required for plasma membrane permeabilization. Engagement of PIP$_2$ is mediated by NaD1 dimers that form a distinctive PIP$_2$-binding 'cationic grip' that interacts with the head groups of two PIP$_2$ molecules. Functional assays using NaD1 mutants reveal that the mechanism of membrane permeabilization by NaD1 is likely to be conserved between fungal and mammalian tumor cells. Together, these data lead to a new perspective on the role of ligand binding and oligomer formation of defensins during membrane permeabilization.

## Results

### NaD1 binds phospholipids including phosphatidylinositol 4,5-bisphosphate (PIP$_2$)

To define the molecular basis of NaD1 target cell membrane permeabilization activity, we set out to identify potential ligands for NaD1. Membrane lipids represent an attractive target for NaD1; therefore, we investigated whether NaD1 interacts with cellular lipids using protein–lipid overlay assays based on lipid strips immobilized with 100 pmoles of various biologically active lipids (*Poon et al., 2010*; *Patel et al., 2013*). NaD1 specifically bound to certain phospholipids, including several phosphatidylinositol mono-/bis-/tri-phosphates, phosphatidylserine, phosphatidic acid, cardiolipin, and sulfatide (*Figure 1A*). Interestingly, NaD1 bound the functionally important plasma membrane phospholipid PIP$_2$ (*Figure 1A*) but did not bind to a panel of other membrane lipids or sphingolipids. To confirm that the ability of NaD1 to engage PIP$_2$ was not a result of immobilization on the lipid strip, we confirmed that NaD1 also bound PIP$_2$ in the context of a membrane bilayer using a liposome pull-down assay (*Figure 1B*).

### PIP$_2$ binding to NaD1 leads to the formation of an arch-shaped oligomer

To gain insight into the NaD1:PIP$_2$ interaction at the atomic level, we determined the crystal structure of NaD1 in complex with PIP$_2$. The structure of monomeric NaD1 (*Lay et al., 2012*) was used to solve the structure of a NaD1:PIP$_2$ complex by molecular replacement and refined to a resolution of 1.6 Å with values of R$_{work}$/R$_{free}$ of 0.155/0.184 (*Table 1*). Upon PIP$_2$ binding, NaD1 forms an arch composed of 14 NaD1 molecules (*Figure 2A*), with a final arch diameter of 90 Å and a width of 35 Å. The asymmetric unit contains all 14 NaD1 molecules that form the final arch, with the symmetry of the arch being entirely non-crystallographic. Fourteen PIP$_2$ molecules are bound in an extended binding groove (*Figure 2B*) on the inside of the arch (*Figure 2A*). The entire oligomeric complex is held together by a complex network of interactions, which include numerous NaD1:NaD1 (*Figure 3A,B*) and NaD1:PIP$_2$ interactions (*Figure 3C,D*). Notably, the arch-shaped oligomer displays a small degree of pitch, which although noticeable is not sufficient to allow the formation of an extended coil in the crystal (*Figure 2B*).

### The NaD1:PIP$_2$ oligomer contains two distinct NaD1:NaD1 interfaces

The observed NaD1:PIP$_2$ oligomer can be described as an assembly of seven NaD1 dimers, which comprise two distinct NaD1:NaD1 interfaces. The first interface is formed by an antiparallel alignment of the β1-strand from each of two NaD1 molecules (monomers I and II) and exhibits two-fold symmetry

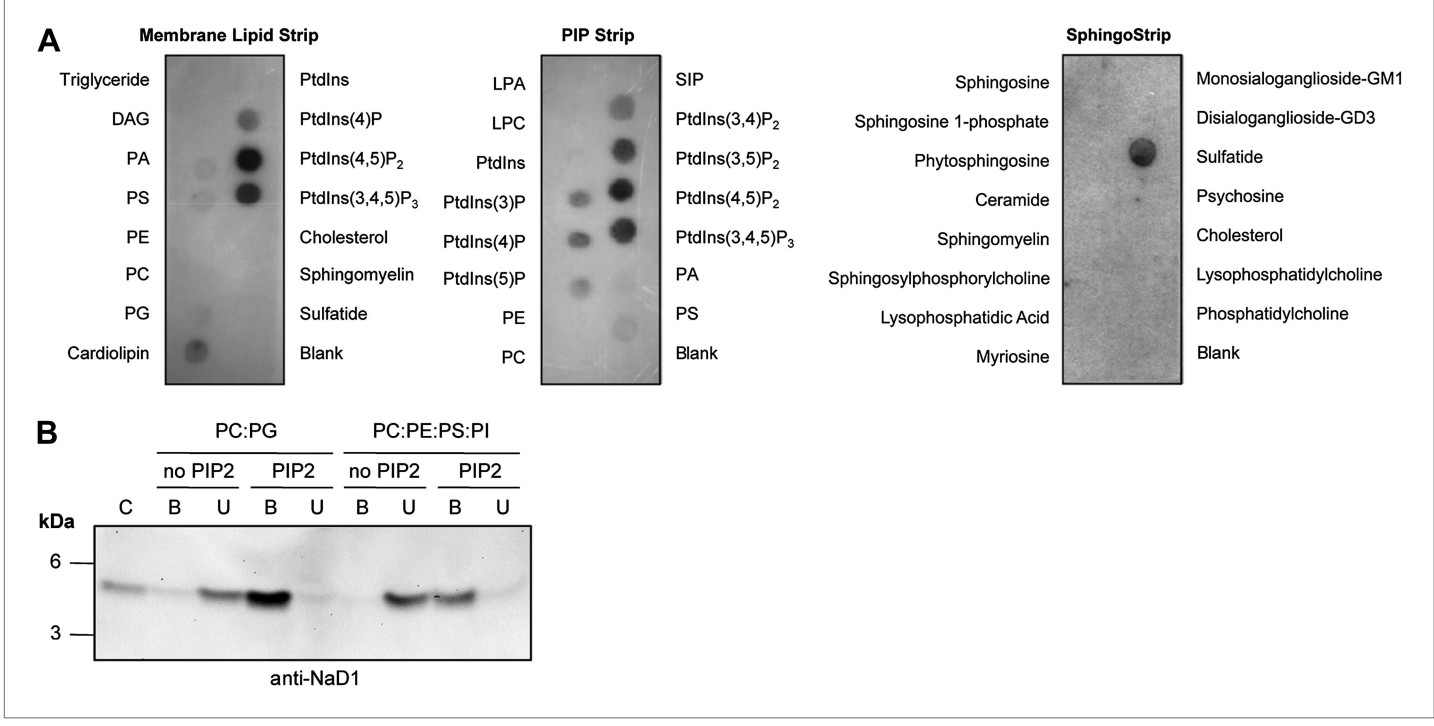

**Figure 1**. Interaction of NaD1 with lipids. (**A**) Detection of NaD1 binding to cellular lipids by protein-lipid overlay assay. Blots are representative of at least two independent experiments for each strip. (**B**) Binding of NaD1 to PIP$_2$-containing liposomes. NaD1 in **A** and **B** was detected using a rabbit anti-NaD1 antibody.

The following figure supplements are available for figure 1:

**Figure supplement 1**. Relative binding of NaD1 to lipids.

between the associated monomers (*Figure 3A*). It comprises an average buried surface area of 430 Å$^2$ and is formed by a network of six hydrogen bonds involving R1, K4, E6, E27, K45, and C47. This dimeric arrangement leads to the formation of a 'cationic grip' (*Figure 4A,B*), which is able to accommodate two PIP$_2$ head groups simultaneously (*Figure 3—figure supplement 1*). A second interface is formed by the dimeric NaD1 (comprising monomers I and II) and adjacent NaD1 monomers III and IV (*Figure 3B*). This interface is formed by hydrogen bonds involving N8 of monomer I, R1, E2, K17; D31 of monomer II, R1, K17; D31 of monomer III; and N8 of monomer IV, effectively forming a dimer of dimers (*Figure 3—figure supplement 1*). The interactions between two dimers are repeated seven times to allow formation of the observed 14-mer. The full 14-mer is thus constructed using two different interfaces.

## PIP$_2$ is bound in an extended binding groove

In addition to NaD1:NaD1 interactions, oligomer formation requires the presence of PIP$_2$. NaD1 binds PIP$_2$ primarily via a 'cationic grip' that is created by a NaD1 dimer, which results in the formation of a distinct binding site (*Figure 3—figure supplement 2*) formed by K4 together with residues 33–40, which comprise a characteristic 'KILRR' motif (*Figure 3C,D*). PIP$_2$ forms a dense network of hydrogen bonds involving K4, H33, K36, I37, L38, and R40 of a single NaD1 monomer. In oligomeric NaD1:PIP$_2$, a single PIP$_2$ binding site also contains interactions with neighboring NaD1 monomers (*Figure 3C,D*; *Figure 3—figure supplement 1*). Bound PIP$_2$ forms additional hydrogen bonds with R40 from monomer II and K36 from monomer IV', with the full PIP$_2$ binding site in the oligomer comprising contributions from three different NaD1 molecules (*Figure 3C,D*). Consequently, oligomer formation appears to be highly cooperative, with multiple interactions between adjacent NaD1 and PIP$_2$ molecules required to form the observed 14-mer (*Figure 3*).

## NaD1:PIP$_2$ oligomers form readily in solution

To confirm that oligomer formation is not a crystallization artifact, we treated mixtures of NaD1 and PIP$_2$ in aqueous solution with the crosslinker BS$^3$, which resulted in covalent cross-linking of multiple NaD1 molecules that occurred only in the presence of PIP$_2$ (*Figure 5A*), whereas NaD1 on its own only

**Table 1.** Data collection and refinement statistics

| | NaD1:PIP₂ native |
|---|---|
| Data collection | |
| Space group | C222₁ |
| Cell dimensions | |
| $a$, $b$, $c$ (Å) | 79.64, 132.04, 153.01 |
| α, β, γ (°) | 90.00, 90.00, 90.00 |
| Wavelength (Å) | 0.9537 |
| Resolution (Å)* | 40.84–1.6 (1.69–1.60) |
| $R_{sym}$ or $R_{merge}$* | 0.092 (0.617) |
| $I/\sigma I$* | 11.6 (2.2) |
| Completeness (%)* | 99.7 (94.7) |
| Redundancy* | 6.7 (5.4) |
| Refinement | |
| Resolution (Å) | 40.37–1.6 |
| No. reflections | 105745 |
| $R_{work}/R_{free}$ | 0.155/0.184 |
| No. atoms | |
| Protein | 10326 |
| Ligand/ion | 845 |
| Water | 816 |
| B-factors | |
| Protein | 21.5 |
| Ligand/ion | 28.9 |
| Water | 31.2 |
| R.m.s. deviations | |
| Bond lengths (Å) | 0.010 |
| Bond angles (°) | 1.651 |

*Values in parentheses are for highest resolution shell.

formed a dimer as reported previously (*Lay et al., 2012*).

## NaD1:PIP₂ forms fibrils

We next imaged NaD1:PIP₂ oligomers using transmission electron microscopy (TEM). Complexes of NaD1:PIP₂ (1:1.2 molar ratio) were applied to a carbon-coated copper grid and imaged. Strikingly, long string-like fibrillar structures were observed when both NaD1 and PIP₂ were present, whereas they were absent on grids bearing either NaD1 or PIP₂ in isolation (*Figure 5B*). Although the NaD1:PIP₂ oligomer we observed by crystallography displays a subtle pitch, it is not sufficient to allow continuous addition or concatenation of 14-mers to form the fibrils observed by TEM, with the ends of two 14-mers running into each other. However, given that the crystal structure of the oligomer reveal an outer diameter of 90 Å, with a corresponding diameter of the fibrils under TEM of 10 nM, additional twisting of the 14-mer could allow for the formation of continuous coils with a diameter to match the fibrils observed under TEM.

## PIP₂ binding and oligomerization of NaD1 are critical for fungal cell killing

Based on our oligomeric NaD1:PIP₂ structure, we performed site-directed mutagenesis on NaD1 to confirm the role of proposed key amino acid residues in PIP₂ binding, oligomerization, and fungal cell killing. Examination of the PIP₂ binding pockets in the NaD1:PIP2 oligomer suggests that R40, which contacts two adjacent PIP₂ molecules simultaneously and interacts with the phosphate moiety at position 4, is critical for cooperative binding of PIP₂ and therefore formation of the NaD1:PIP₂ oligomer (*Figure 6A*). Mutation of R40 should not lead to loss of PIP₂ binding, since PIP₂ would still form five hydrogen bonds and ionic interactions with NaD1 and should only impact oligomerization. In contrast, I37 contributes to PIP₂ binding but not oligomerization. We generated recombinant proteins of NaD1 (rNaD1) and NaD1 mutants (rNaD1(R40E) and rNaD1(I37F)) and confirmed their correct folding by CD spectroscopy (data not shown) and evaluated the ability of the mutant NaD1 to bind phospholipids, undergo PIP₂-induced oligomerization, and kill the filamentous fungus *F. oxysporum* f. sp. vasinfectum.

As predicted, mutation of R40 to glutamic acid led to a largely unchanged binding to PIP₂, with the remaining five hydrogen bonds and ionic interactions formed between PIP₂ and rNaD1(R40E) compensating for the loss of two ionic interactions as well as the charge repulsion. However, it did result in reduced binding to PI(4)P (*Figure 6B*) and oligomerization (*Figure 6C*) that correlated with substantially reduced fungal cell killing (*Figure 6D*). It is important to note that although PIP₂ binding was maintained, the loss of interaction with the 4-phosphate moiety of PIP₂ results in loss of cooperative binding and therefore ablation of oligomerization, which is critically dependent on R40 forming 'bridging' interactions between two neighboring PIP₂ molecules. In contrast, mutation of I37 to phenylalanine had little effect on PIP₂ binding specificity, oligomerization, and fungal cell killing (*Figure 6B–D*). These data support our defined NaD1-PIP₂ structure and demonstrate that the coordinated oligomerization of NaD1 by interaction with PIP₂ is an important event during fungal cell killing.

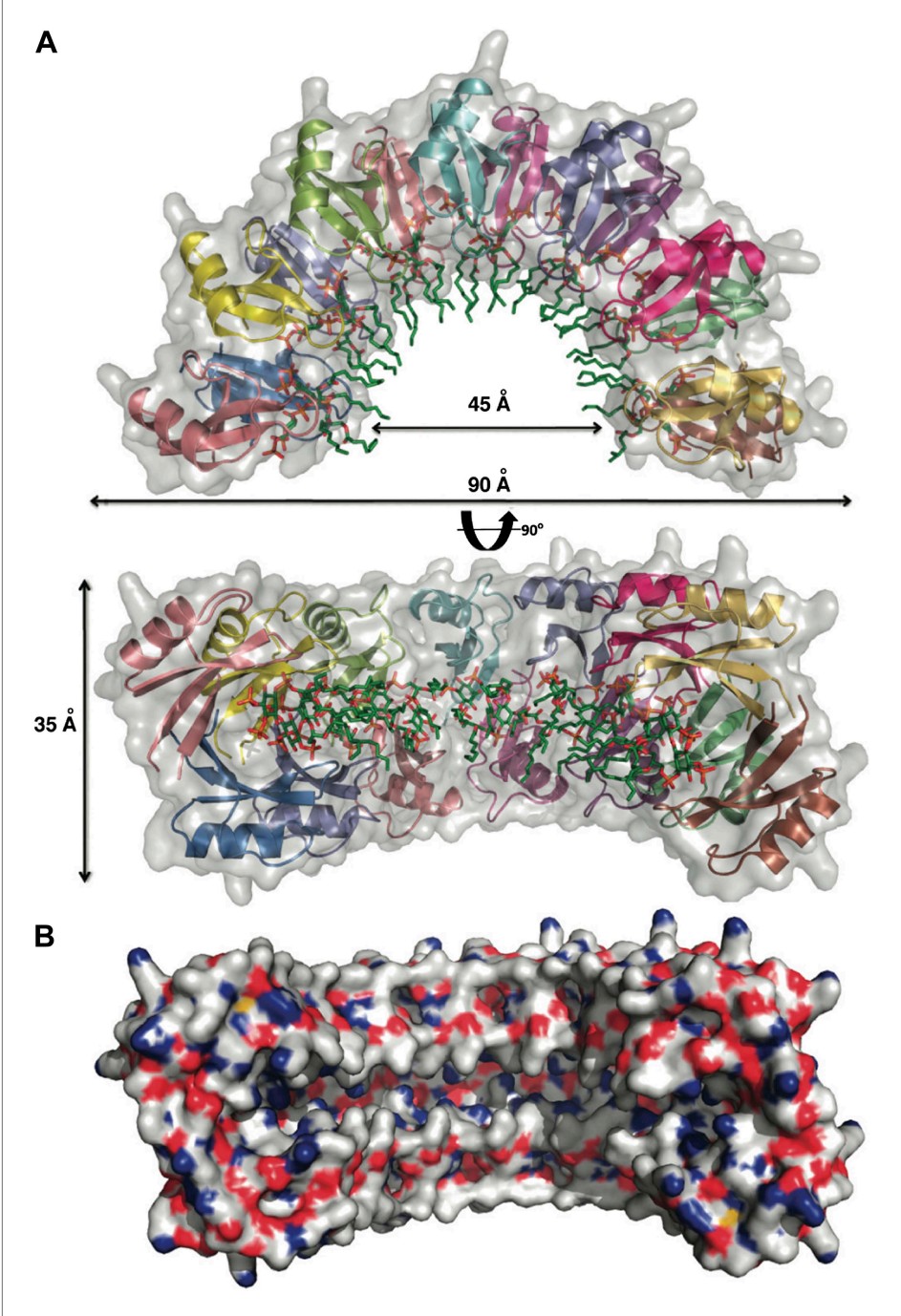

**Figure 2**. Crystal structure of the NaD1:PIP$_2$ complex. (**A**) Two orthogonal views of a cartoon representation of the NaD1:PIP$_2$ oligomer comprising 14 NaD1 monomers (shown as ribbons) and 14 PIP$_2$ molecules (shown as green sticks). The surface of the NaD1 oligomer is shown in gray. (**B**) Surface representation of the NaD1 14-mer, displaying the extended binding groove on the inside of the arch. Coloring is by atom type (N in blue, O in red, S in yellow, and C in gray). For clarity the 14 bound PIP$_2$ molecules were omitted.

## NaD1 permeabilizes the plasma membrane of mammalian tumor cells

Since PIP$_2$ is a critical component of mammalian plasma membranes, we investigated whether NaD1 also harbored permeabilization activity against mammalian cells. To this end, we performed a flow

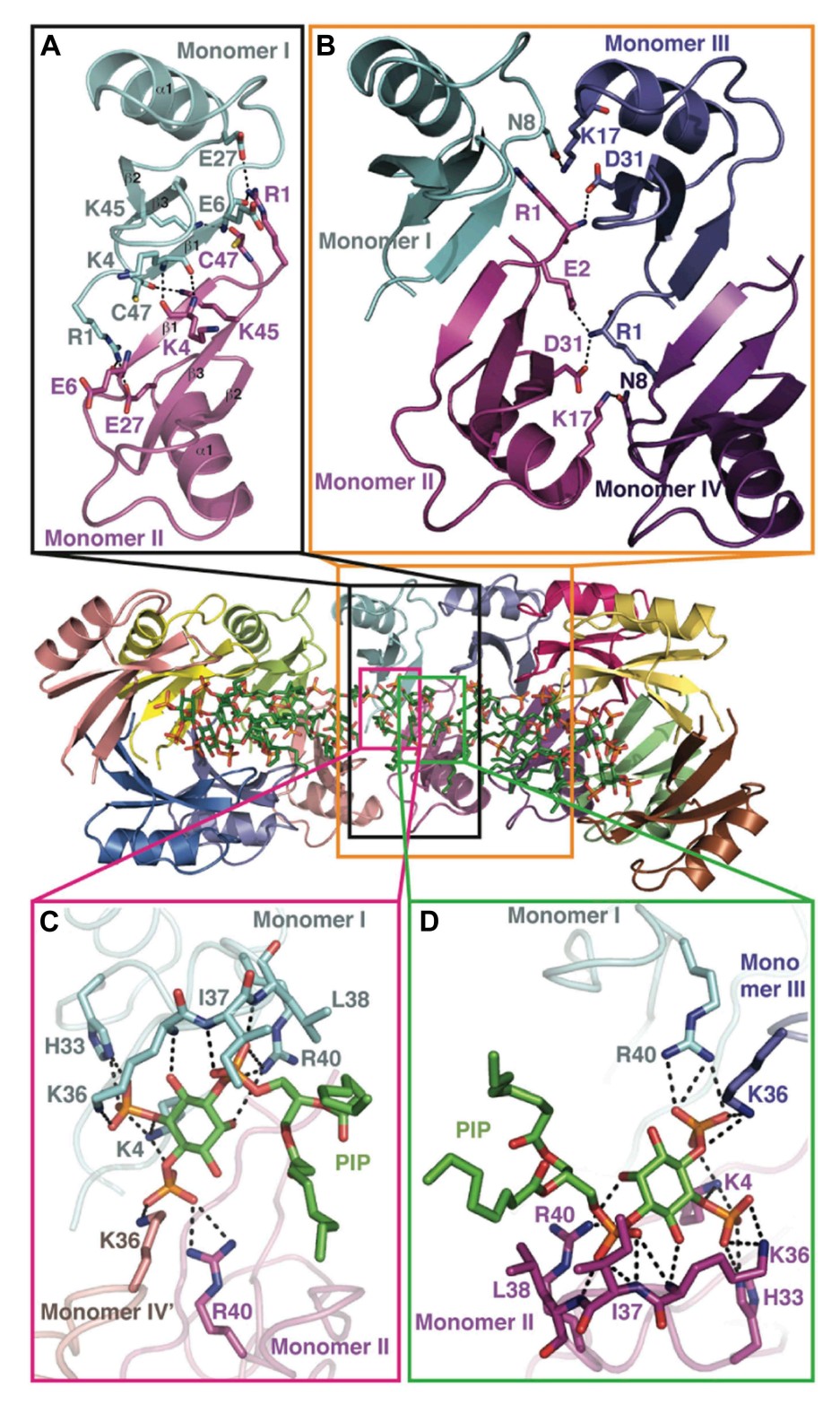

**Figure 3**. Detailed view of the crystal structure of the NaD1:PIP$_2$ complex. In all panels, hydrogen bonds and salt bridges are shown as black dotted lines. (**A**) View of the interface of two NaD1 monomers revealing the hydrogen bonding pattern, with monomer I shown in cyan and monomer II in magenta. Secondary structure elements are labeled in black. For clarity bound PIP$_2$ molecules are omitted. (**B**) Cartoon diagram of four molecules of NaD1 forming a dimer of dimers.
*Figure 3. Continued on next page*

*Figure 3. Continued*

(**C**) PIP$_2$ binding site on monomer I. Cartoon diagram of the PIP$_2$ binding site in monomer I on dimeric NaD1. (**D**) PIP$_2$ binding site on monomer II. Cartoon diagram of the PIP$_2$ binding site on monomer II on dimeric NaD1.
The following figure supplements are available for figure 3:

**Figure supplement 1**. Cartoon of two NaD1 dimers with four bound PIP$_2$ molecules.

**Figure supplement 2**. Simulated anneal omit map of a single PIP$_2$ molecule bound to an NaD1 dimer, contoured at 1σ.

cytometry-based cell permeabilization assay on U937 monocytic lymphoma cells to measure uptake of the membrane-impermeable nucleic acid dye propidium iodide (PI). NaD1 permeabilized the plasma membrane of the U937 cells and induced a change in cell morphology in a concentration-dependent

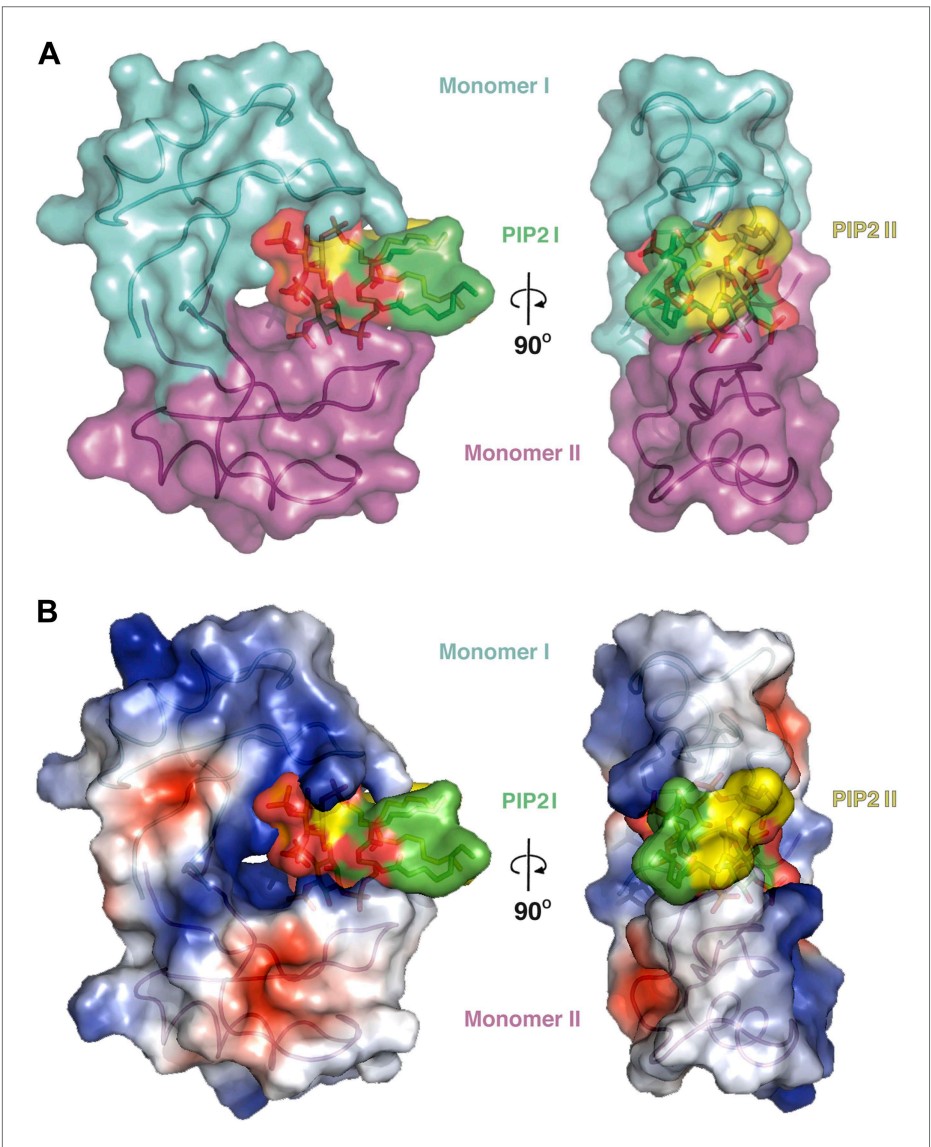

**Figure 4**. The dimeric NaD1 'cationic grip' with two bound PIP$_2$ molecules. (**A**) Surface view in two orientations of a NaD1 dimer (monomer I in cyan and monomer II in magenta) with two bound PIP$_2$ molecules (yellow and green). (**B**) The same as in **A** except that the surface shows a qualitative electrostatic representation (blue is positive, red in negative, and white is uncharged or hydrophobic). Figure generated using Pymol.

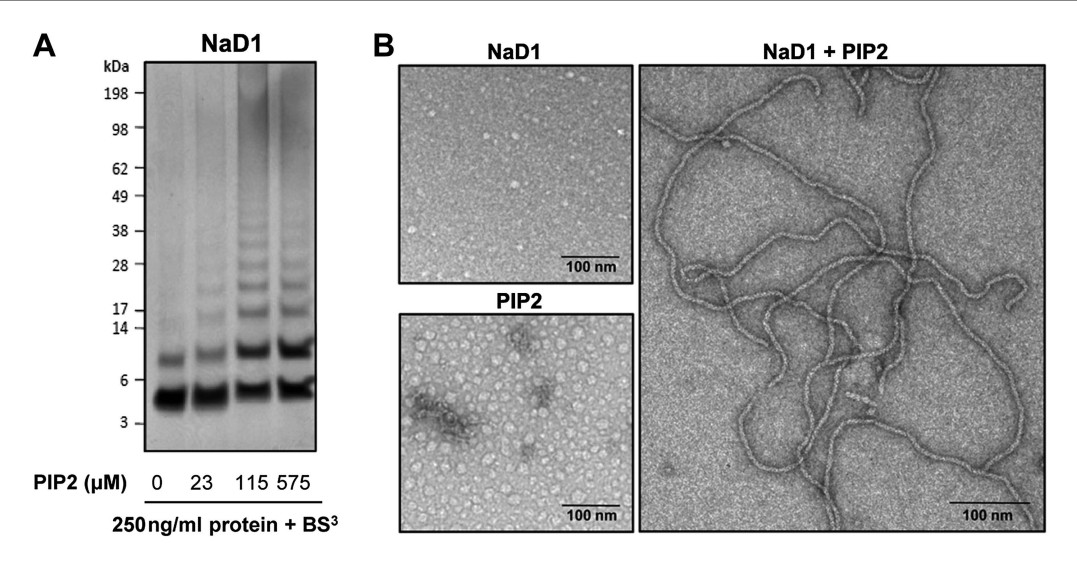

**Figure 5**. NaD1 forms oligomers with PIP$_2$. (**A**) Ability of NaD1 to form multimers in the presence of PIP$_2$ as determined by protein–protein cross-linking with BS$^3$ followed by SDS-PAGE and Coomassie Brilliant Blue staining. (**B**) TEM of NaD1:PIP$_2$ complexes. TEM micrographs of NaD1 alone, PIP$_2$ alone, or NaD1 in complex with PIP$_2$. Data in **A** and **B** are representative of at least two independent experiments.

manner (**Figure 7A**). Furthermore, rapid leakage of intracellular ATP from U937 cells was observed within the first 200 s following exposure to NaD1 (**Figure 7—figure supplement 1**).

NaD1-mediated lysis of U937 cells was confirmed by the uptake of FITC-dextran (up to 40 kDa) and the release of lactate dehydrogenase (140 kDa) into the supernatant after NaD1 treatment (**Figure 7B,C**). In contrast, reduced and alkylated NaD1 (NaD1$_{R\&A}$) showed no cytotoxic activities against U937 cells (**Figure 7—figure supplement 2**), confirming the importance of the NaD1 tertiary structure for the ability to induce membrane permeabilization. It should be noted that NaD1 also permeabilized a diverse range of normal primary human cells and tumor cell lines (**Figure 7—figure supplement 3**), with the highest levels of activity exhibited against tumor cell lines. Collectively, these data suggest that, in addition to antifungal activity, NaD1 also exhibits antiproliferative properties against mammalian cells.

We then sought to examine changes in cell morphology upon NaD1 treatment. Live confocal laser scanning microscopy (CLSM) revealed rapid changes on the cell surface of NaD1-permeabilized tumor cells and showed the formation of large plasma membrane blebs, with adherent cells (HeLa and PC3) forming multiple blebs of different sizes (**Video 1**) and non-adherent cells (U937) forming typically one to two large blebs (**Video 2**; **Figure 8A**). Moreover, bleb size was frequently larger than the actual cell (diameter >20 μm) and did not retract over a period of 20 min (**Figure 8—figure supplement 1**).

In our CLSM studies, we noticed that NaD1-induced membrane blebbing typically coincided with PI uptake (**Video 1**). To determine whether membrane blebbing occurs prior to, during, or following membrane permeabilization, we treated U937 cells with NaD1 in the presence of PI and 4 kDa FITC-dextran to monitor the entry of these molecules into NaD1-sensitive cells (**Figure 8B**; **Video 3**). FITC-dextran and NaD1 were added at 00:35 min, with FITC-dextran being excluded from cells with an intact membrane. Bleb formation was first observed for the cell located at the center of the panel at 03:25 min, with PI staining appearing at a specific point at the edge of the bleb. From 03:25 to 04:15 min, PI staining was observed in the bleb and the cytoplasm with FITC-dextran also entering the cell from the bleb site. At 04:20 min, PI-stained molecules were 'expelled' out of the cell, possibly at the same region that PI first entered the bleb. Similar results were also observed for PC3 cells (**Figure 8—figure supplement 2**). These data suggest that (i) small molecules such as PI can enter the cell initially at a 'weakened' point at the membrane bleb, (ii) the bleb continues to enlarge while PI and 4 kDa FITC-dextran enters, and (iii) intracellular contents are released at the bleb site, representing cytolysis.

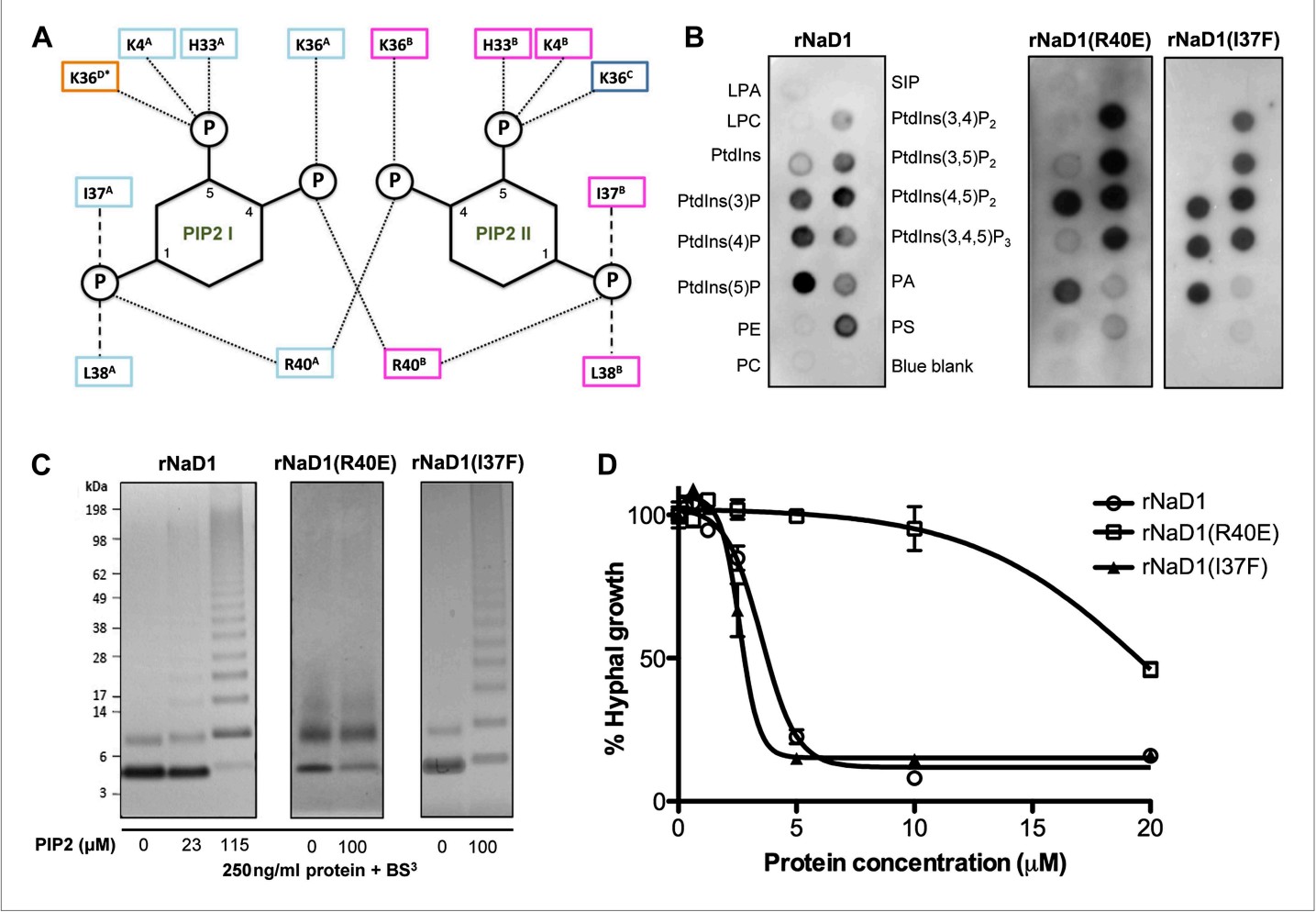

**Figure 6**. Multimerization of the NaD1:PIP$_2$ complex. (**A**) Schematic representation of residues from neighboring NaD1 monomers involved in binding two PIP$_2$ molecules. Ability of rNaD1, rNaD1(R40E), and rNaD1(I37F) to (**B**) bind cellular lipids by protein-lipid overlay assay, (**C**) form multimers in the presence of PIP$_2$ as determined by protein–protein cross-linking with BS$^3$ followed by SDS-PAGE and Coomassie Brilliant Blue staining, and (**D**) to inhibit fungal cell growth. Error bars in **D** indicate SEM (n = 3). Data in **B**–**D** are representative of at least two independent experiments.

The following figure supplements are available for figure 6:

**Figure supplement 1**. Relative binding of rNaD1, rNaD1(I37F), and rNaD1(R40E) to lipids.

## NaD1 interacts with phosphoinositides in cellular membranes of tumor cells

We next determined the specific mechanism by which NaD1 permeabilizes mammalian cells. Firstly, we tested the binding of BODIPY-labeled NaD1 to U937 cells. BODIPY-NaD1 permeabilized U937 cells at a level comparable to unlabeled NaD1 and bound to both viable (7AAD-negative) and permeabilized (7AAD-positive) cells, with more BODIPY-NaD1 bound to membrane-damaged cells (*Figure 9A*). These data suggest that NaD1 can interact with U937 cells prior to membrane permeabilization and accumulates on/within NaD1-sensitive cells.

We then determined the subcellular localization of BODIPY-NaD1 on permeabilized tumor cells. BODIPY-NaD1 accumulated at membrane bleb(s), in the cytoplasm and nucleolus and possibly at certain cytoplasmic organelles in U937, PC3, and HeLa cells (*Figure 9B*; *Videos 4 and 5*). It is worth noting that the formation of large plasma membrane blebs has been reported previously in mammalian cells in which physically- or chemically-induced detachment of the plasma membrane from the actin cortex had occurred, including through enzymatic modification or sequestration of the inner membrane phospholipid, PIP$_2$ (*Niebuhr et al., 2002*; *Sheetz et al., 2006*; *Keller et al., 2009*).

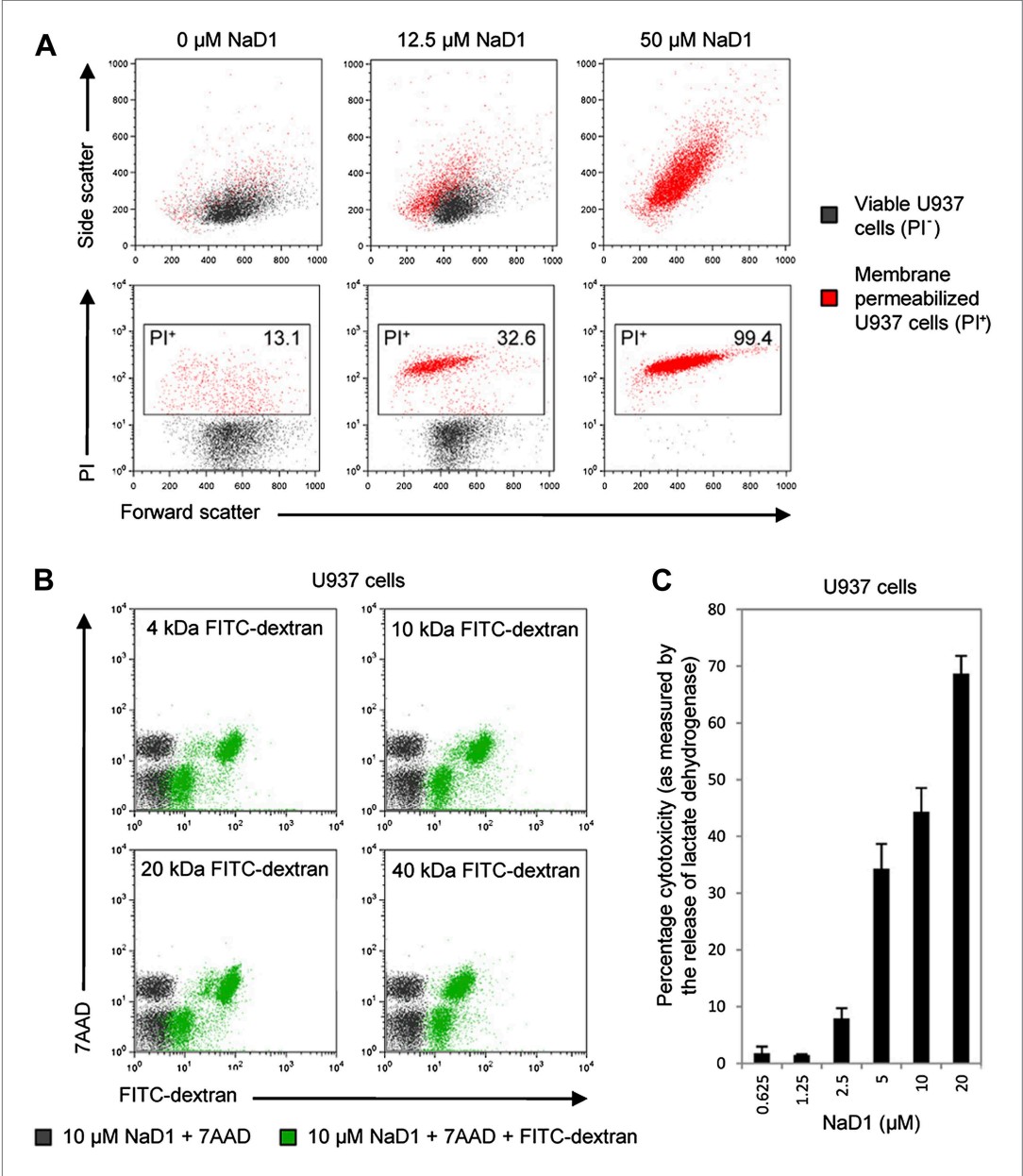

**Figure 7**. NaD1 kills mammalian tumor cells by membrane permeabilization. (**A**) Forward scatter, side scatter, and PI uptake analysis of U937 cells treated with NaD1. (**B**) Binding of FITC-dextran and (**C**) LDH release by NaD1-treated U937 cells. Error bars in **C** indicate SEM (n = 3). Data in **A**–**C** are representative of at least two independent experiments.

The following figure supplements are available for figure 7:

**Figure supplement 1**. NaD1 rapidly permeabilizes U937 cells.

**Figure supplement 2**. Reduced and alkylated NaD1 (NaD1$_{R\&A}$) does not permeabilize U937 cells.

**Figure supplement 3**. Tumor/transformed cells are more susceptible to killing by NaD1 than normal/primary cells.

We then asked whether the binding of NaD1 to PIP$_2$ at the inner leaflet of the plasma membrane could lead to the formation of blebs, as PIP$_2$ is a key mediator of cytoskeleton-membrane interactions (***Raucher et al., 2000***; ***Sheetz, 2001***). HeLa cells expressing GFP-PH(PLCδ), which binds specifically to

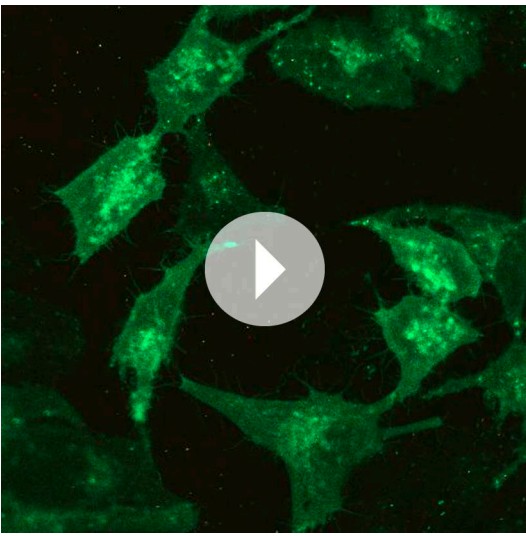

**Video 1**. NaD1 rapidly induces membrane blebbing and permeabilization of HeLa cells. Live CLSM of PKH67-stained HeLa cells in the presence of PI. Cells were imaged over a period of 10 min (5 s/frame), with NaD1 (10 µM) being added to cells at 1 min.

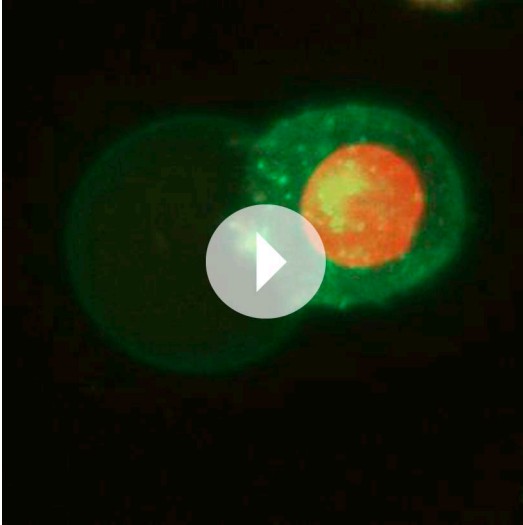

**Video 2**. Formation of a single large membrane bleb on a U937 cell following NaD1 treatment. Three-dimensional reconstruction of CLSM images of a NaD1-treated (10 µM) PKH67-/PI-stained U937 cell.

$PIP_2$, were treated with NaD1 and showed a marked delay from the initiation of blebbing (rapid small membrane blebbing) to membrane permeabilization compared with cells expressing free GFP (*Figure 10*; *Videos 6 and 7*). Quantitation of the kinetics of cell permeabilization indicated that GFP-PH(PLCδ)-expressing cells took approximately 2.5 times as long as GFP only cells to permeabilize in response to NaD1 (235 ± 40 s vs 90 ± 20 s, respectively) (*Figure 10B*). These results suggest that the expression of GFP-PH(PLCδ) may compete with NaD1 for $PIP_2$ binding at the inner leaflet of the plasma membrane and interfere with NaD1-induced cell permeabilization.

We then evaluated the effect of our loss-of-function mutant rNaD1(R40E), which exhibited reduced killing activity of fungal cells, on U937 cells. rNaD1(R40E) showed dramatically reduced ability to permeabilize these tumor cells, in contrast to the control rNaD1 and rNaD1(I37F) proteins as demonstrated by the PI uptake assay (*Figure 11A*). Similarly, rNaD1(R40E)-treated U937 cells displayed no LDH release (*Figure 11B*) or FITC-dextran uptake (*Figure 11C*) compared to the control proteins.

In their totality, these data support our notion that the coordinated oligomerization of NaD1 by interaction with $PIP_2$ is a critical event for fungal and tumor cell killing.

## Discussion

CAPs, of which defensins are a major family, are weapons within the armory of host defense peptides that are utilized by animals and plants in their fight against pathogenic threats. NaD1 is a defensin from the ornamental tobacco that has potent activity against fungi and yeast (*Lay et al., 2003a, 2012*; *van der Weerden et al., 2008, 2010*; *Hayes et al., 2013*). This defensin operates by a three-step mechanism—specific interaction with the cell wall followed by permeabilization of the plasma membrane and entry into the cytoplasm (*van der Weerden et al., 2010*). However, the precise molecular basis of membrane permeabilization and passage through the membrane is poorly defined, particularly in terms of the specific lipid targets on the membrane and the structural definition of defensin-lipid interactions.

In this study, we identified the lipid targets of NaD1 as phosphoinositides (PIPs) and show that the binding of particular PIPs such as $PIP_2$, mediates NaD1 oligomerization and membrane permeabilization. This expands on our previous study revealing that the ability of NaD1 to homo-dimerize enhances its antifungal activity (*Lay et al., 2012*). It has been postulated that the ability of human defensins and other CAPs to form higher oligomeric states at the plasma membranes of target cells is a contributing factor in membrane disruption and/or permeabilization (*Hill et al., 1991*; *Wimley et al., 1994*; *Hoover et al., 2000*; *Mader and Hoskin, 2006*; *Wei et al., 2010*), but to date no structural explanations have been reported. Our crystal structure of a NaD1:$PIP_2$ oligomeric

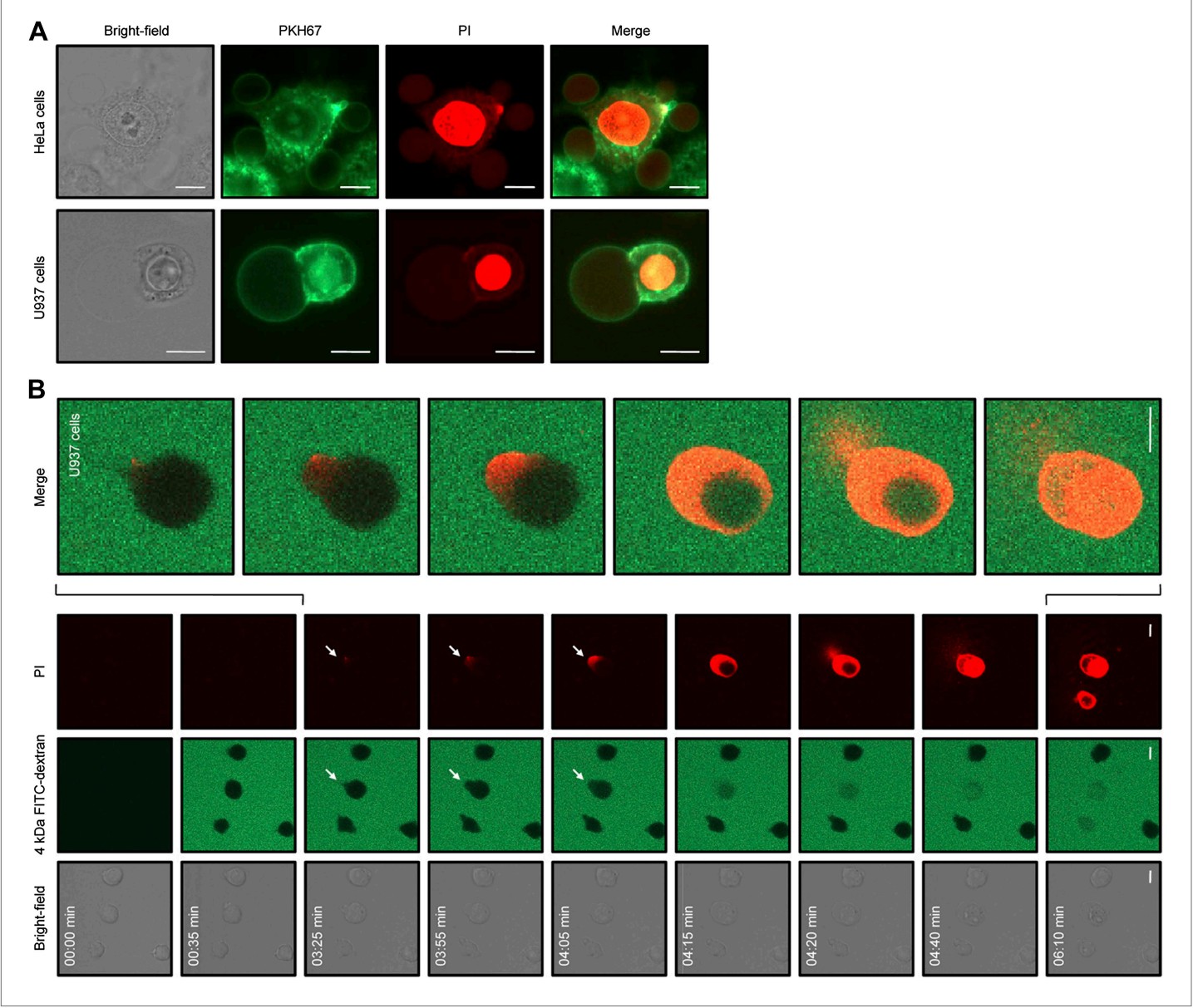

**Figure 8**. NaD1 induces membrane blebbing of tumor cells. (**A**) CLSM of PKH67-stained NaD1 (10 µM) permeabilized HeLa and U937 cells. (**B**) CLSM of U937 cells treated with NaD1 (20 µM) in the presence of PI and 4 kDa FITC-dextran. Arrows indicate entry of PI. It should be noted that in this experiment the detector gain on the helium–neon laser (red channel) was increased compared to that used in **A** to enable visualization of the cell lysis events. Scale bars represent 10 µm. Data in **A** and **B** are representative of at least two independent experiments.

The following figure supplements are available for figure 8:

**Figure supplement 1**. NaD1-induced membrane blebs do not retract once U937 cells are permeabilized.

**Figure supplement 2**. NaD1-mediated membrane permeabilization occurs at the blebs of PC3 cells.

complex reveals the first detailed molecular description of a plant defensin-lipid interaction and identifies a new mechanism of membrane permeabilization.

The proposed antifungal mechanisms of action for plant defensins are diverse and includes membrane permeabilization, generation of reactive oxygen species with induction of apoptosis, and dysregulation of $Ca^{2+}$ influx and $K^+$ efflux (reviewed in *De Coninck et al., 2013*; *van der Weerden and Anderson, 2013*). The structural basis of the interaction of these defensins with lipid has not been

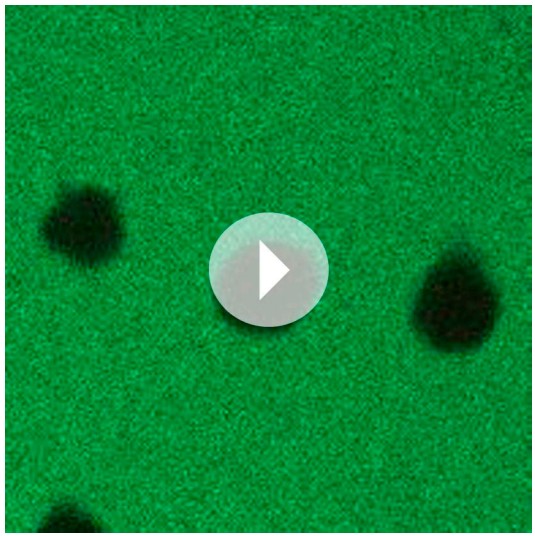

**Video 3**. NaD1-mediated membrane permeabilization occurs at the blebs of U937 cells. Live CLSM of U937 cells treated with NaD1 in the presence of PI and 4 kDa FITC-dextran. Cells were imaged over a period of 10 min (5 s/frame), with NaD1 (20 µM) and 4 kDa FITC-dextran (100 µg/ml) being added to cells at 30 s.

defined. However, it is interesting to note that the equivalent region to the 'KILRR' loop of NaD1 (the loop between the β2 and β3 strands), that is critical in forming the lipid-binding 'cationic grip', has been implicated as functionally important for the antifungal activity of a number of plant defensins. For example, mutations in the equivalent region of RsAFP2 abolished antifungal activity (*De Samblanx et al., 1997*), and a chimeric protein generated by replacing this region of MsDef1 with the equivalent region of MtDef4 (a functionally distinct defensin that does not bind sphingolipid) resulted in functional conversion into a defensin able to inhibit the growth of a glucosylceramide-deficient, MsDef1-resistant *F. graminearum* strain (*Sagaram et al., 2011*). Based on our observation that NaD1 binds phosphoinositides through the same loop region, it is tempting to speculate that plants have evolved a suite of defensins that specifically interact through their β2–β3 loop regions with different membrane lipids to mediate fungal cell killing. The β2–β3 region has also been implicated in the α-amylase activity of the plant defensin, VrD1, suggesting that this region is also functionally important in defensins with different activities (*Lin et al., 2007*). Not surprisingly, the

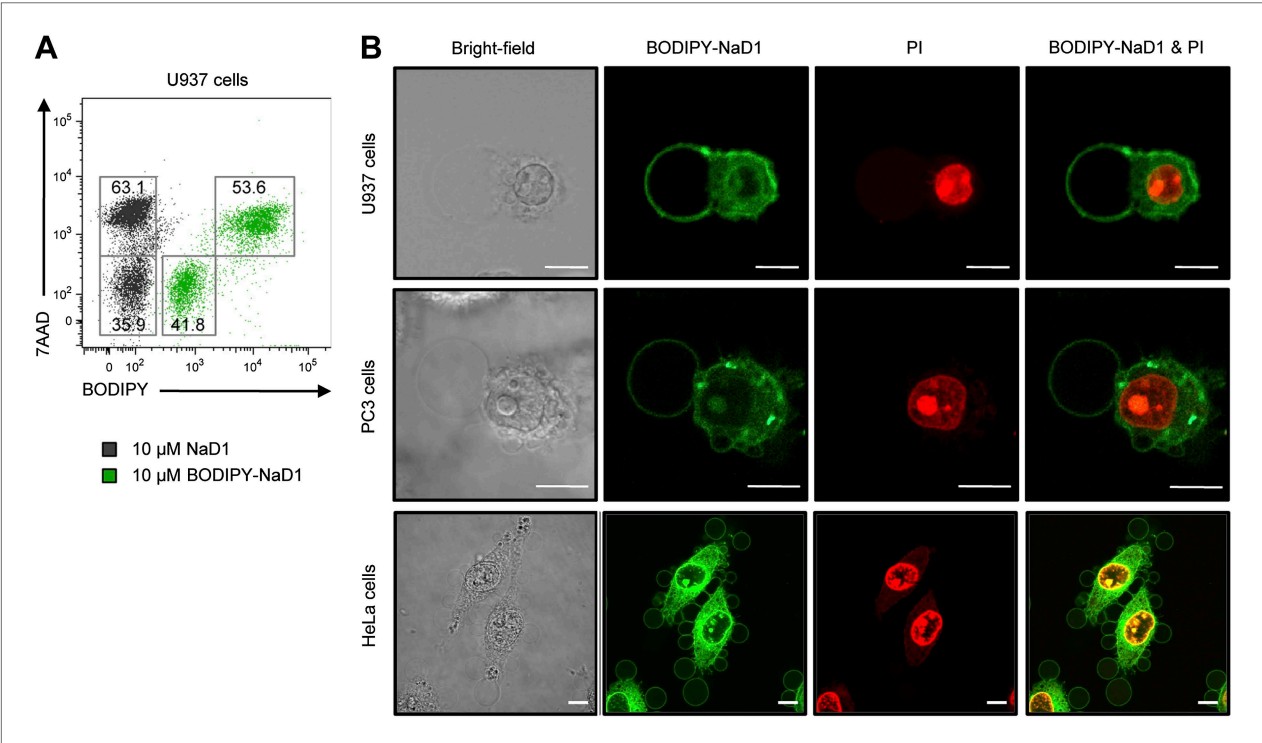

**Figure 9**. Subcellular localisation of BODIPY-NaD1 in tumor cells. (**A**) Detection of BODIPY-NaD1 binding to viable and permeabilized U937 cells by flow cytometry. (**B**) CLSM of subcellular localization of BODIPY-NaD1 (10 µM) on permeabilized U937, PC3, and HeLa cells. Scale bars represent 10 µm. Data in **A** and **B** are representative of at least two independent experiments.

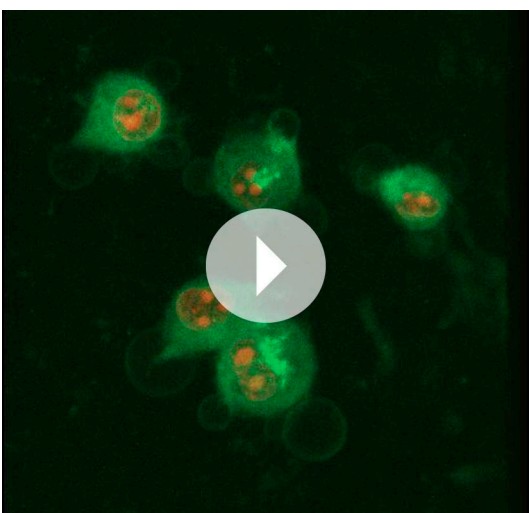

**Video 4**. BODIPY-NaD1 accumulates at the plasma membrane and certain intracellular organelles. Three-dimensional reconstruction of CLSM images of BODIPY-NaD1-treated (10 μM) PI-stained PC3 cells.

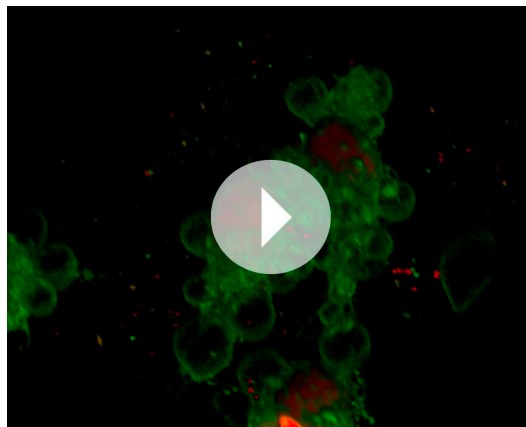

**Video 5**. BODIPY-NaD1 accumulates at the plasma membrane and certain intracellular organelles. Three-dimensional reconstruction of CLSM images of BODIPY-NaD1-treated (10 μM) PI-stained HeLa cells.

β2–β3 region of plant defensins exhibits considerable sequence divergence which is likely to reflect their ability to bind different ligands and therefore the various mechanisms of antifungal activity (*van der Weerden and Anderson, 2013*).

The antifungal plant defensins, DmAMP1 and RsAFP2, interact with different sphingolipids. DmAMP1 binds $M(IP)_2C$ (*Thevissen et al., 2000, 2003*) and *S. cerevisiae* strains that have gene disruptions to encoded proteins within the $M(IP)_2C$ biosynthetic pathway are rendered DmAMP1-resistant (*Thevissen et al., 2005*). In contrast, RsAFP2 binds to GlcCer, present in fungi such as *Pichia pastoris* and *C. albicans* and does not cause permeabilization or growth inhibition of strains that do not express GlcCer (*Thevissen et al., 2004*). The fact that DmAMP1 is able to act on these strains supports the notion that DmAMP1 and RsAFP2 have different lipid targets. It is worthwhile noting that RsAFP2 does not bind to GlcCer derived from human or soybean (*Thevissen et al., 2004*), suggesting that there is some level of selectivity, while its inability to permeabilize artificial liposomes containing GlcCer indicates that binding alone is insufficient for its permeabilization action (*Thevissen et al., 2004*).

This is the first report of a defensin from any species targeting a phosphoinositide, such as $PIP_2$. Interestingly, a number of toxins have been reported to directly or indirectly target $PIP_2$, resulting in plasma membrane reorganization and permeabilization. The marine bacterium *Vibrio parahaemolyticus* causes gastroenteritis in humans by acting as an inositol polyphospholipid 5' phosphatase. It targets $PIP_2$ by catalyzing the removal of the 5' phosphate moiety, resulting in the disruption of $PIP_2$-mediated cytoskeletal interactions leading to membrane blebbing and cell lysis (*Broberg et al., 2010*). A similar mode of action has also been described for the effector protein ipgD of the bacillary dysentery causing Gram-negative pathogen *Shigella flexneri* (*Niebuhr et al., 2002*). The equinatoxin from the sea anemone

*Actinia equina* also causes membrane blebbing and cell lysis via a mechanism involving $Ca^{2+}$-mediated $PIP_2$ hydrolysis at the inner membrane and the formation of membrane pores in target cells (*Garcia-Saez et al., 2011*). In this study, we show that NaD1 also causes plasma membrane blebbing and permeabilization of human cells. However, in contrast to the enzymatic or pore-forming mechanism(s) of the above toxins, it does so through the direct binding of $PIP_2$. The ability of NaD1 to form an oligomeric complex with $PIP_2$ suggests it could potentially sequester $PIP_2$ from the plasma membrane, leading to membrane destabilization, blebbing, and ultimately cell lysis. Such an oligomerization process would be predicted to efficiently displace any $PIP_2$-binding molecules. Indeed, the ability of lipid-binding proteins to oligomerize on membranes has been proposed as an important mechanism in mediating high-avidity membrane interactions (*Lemmon, 2008*).

The increased sensitivity of tumor cells to NaD1 compared with its effects on healthy primary cells may be attributed to a number of differences in the physical properties of the plasma membranes of these two cell types. These include an increase in the expression of negatively charged outer

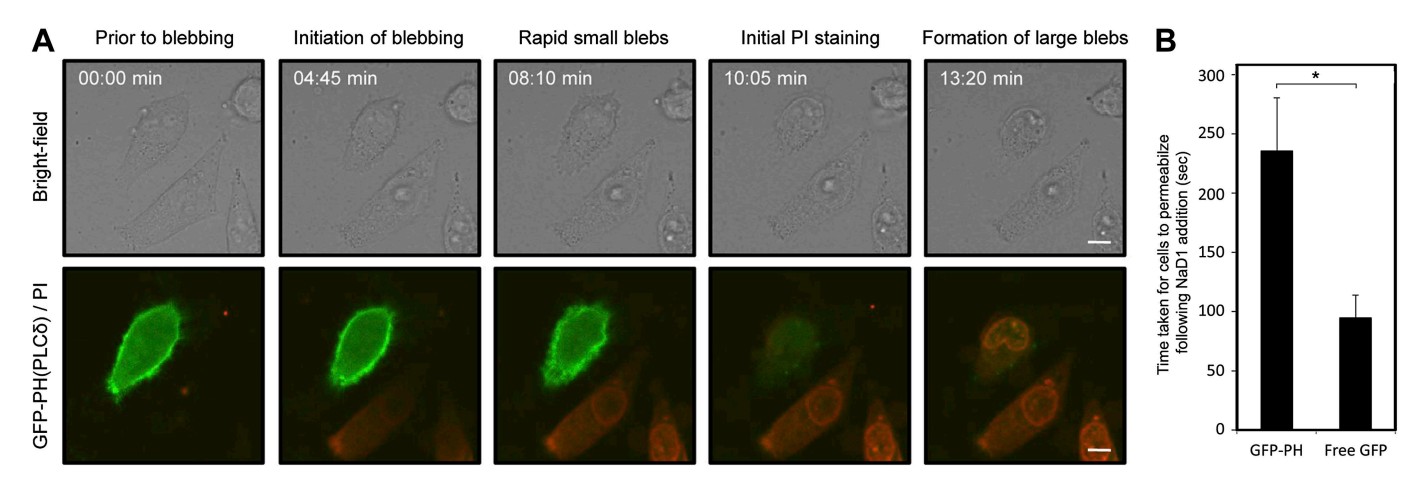

**Figure 10**. Expression of GFP-PH(PLCδ) in HeLa cells significantly delays NaD1-mediated cell permeabilization compared with cells expressing free GFP. (**A**) CLSM of NaD1 (10 μM) treated HeLa cells expressing GFP-PH(PLCδ). Scale bars represent 10 μm. (**B**) The average length of time taken for NaD1 (10 μM) to permeabilize (PI-positive) GFP-PH(PLCδ)-expressing vs free GFP-expressing HeLa cells were analyzed over a period of 15 min. For GFP-PH(PLCδ)-expressing cells, n = 21; for free GFP-expressing cells, n = 29. Error bars indicate SEM, * = p<0.005.

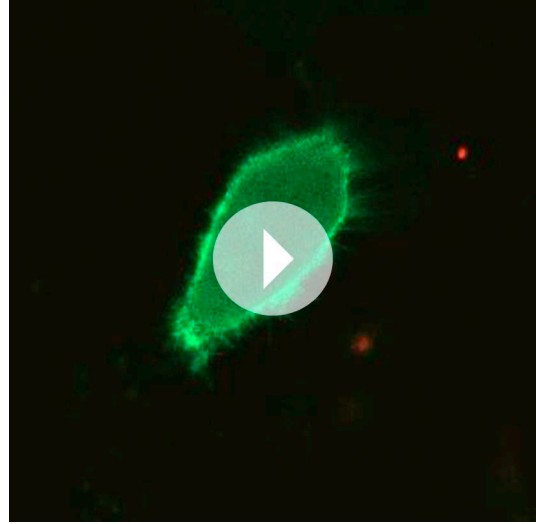

**Video 6**. Expression of GFP-PH(PLCδ) in HeLa cells delays NaD1-mediated cell permeabilization. Live CLSM of GFP-PH(PLCδ) transfected HeLa cells treated with NaD1 in the presence of PI. Cells were imaged over a period of 15 min (5 s/frame), with NaD1 (10 μM) being added to cells at 30 s.

membrane components, such as O-glycosylated mucins (*Yoon et al., 1996*; *Kufe, 2009*) and phosphatidylserine (*Utsugi et al., 1991*; *Ran et al., 2002*), which could allow stronger initial electrostatic interactions between the cationic NaD1 and the cell surface. The increased levels of microvilli (*Chaudhary and Munshi, 1995*; *Ino et al., 2002*) and higher degree of membrane fluidity (*Sok et al., 1999*; *Zeisig et al., 2007*) in tumor cells may also facilitate the aggregation of a greater number of NaD1 molecules to the cell surface and assist in penetration and/or destabilization of the membrane, respectively. The precise mechanism of action for the increased sensitivity of mammalian tumor cells over normal cells to defensins and whether this approach can be harnessed for selective tumor cell killing remains to be determined.

The ability of NaD1 to permeabilize both fungal and mammalian cell membranes suggests a common mode of interaction with membranes and our findings described herein implicate PIP$_2$ in both settings. Although the plasma membranes on mammalian and fungal cells are different in overall lipid compositions (typically mammalian cells are rich in zwitterionic phospholipids whereas fungal membrane have higher levels of anionic phospholipids), PIP$_2$ is important in both species (*Di Paolo and De Camilli, 2006*; *van Meer et al., 2008*). PIP$_2$ is normally found on the inner leaflet of mammalian cells and although it is a minor species comprising only 0.5–1% of phospholipids, it plays a major regulatory role in a number of important membrane-related processes, including signal transduction, ion channel function, and cytoskeletal attachment (*McLaughlin and Murray, 2005*). Similar functions for PIP2 have been reported or suggested in the plasma membrane of fungi (*van Meer et al., 2008*). The importance of PIP2 in many fundamental cellular processes would certainly make it an attractive target for defense against pathogens.

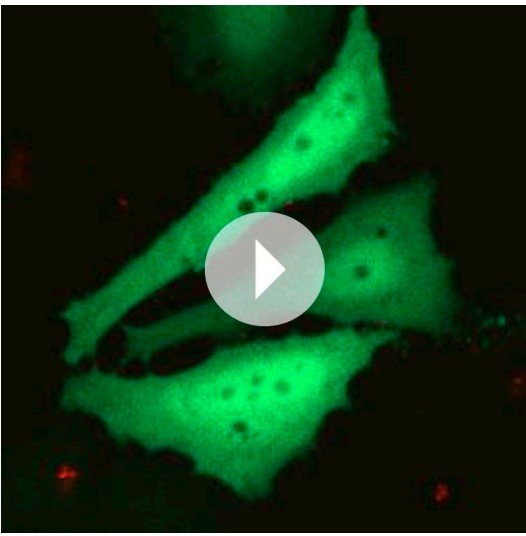

**Video 7**. Expression of free GFP in HeLa cells does not delay NaD1-mediated cell permeabilization. Live CLSM of free GFP transfected HeLa cells treated with NaD1 in the presence of PI. Cells were imaged over a period of 20 min (5 s/frame), with NaD1 (10 μM) being added to cells at 30 s.

In addition to binding $PIP_2$, we show that NaD1 is also able to bind to a number of other phospholipids. The functional consequences of the promiscuous binding by NaD1 remains to be determined. The various phospholipids recognized by NaD1 have different subcellular distributions and functions. In its normal physiological setting as a plant antifungal molecule, the ability of NaD1 to bind a number of different phospholipids may enable defense against a wide array of different fungal pathogens. How NaD1 is able to enter cells remains to be defined. It is possible that the binding of PIPs such as PI(3)P may mediate entry of NaD1 into cells. Indeed, cell surface-expressed PI(3)P has been suggested to mediate entry of eukaryotic pathogen effector molecules, such as Avr from oomycetes and fungi, into plant cells (*Kale et al., 2010*).

Structurally, one can envisage that PIPs other than $PIP_2$ can fit into the cationic grip and be able to induce NaD1 oligomerization. Inspection of the cationic grip indicates that an additional phosphate group at the 3-inositol position can be accommodated and would contact R40 in a manner similar to the phosphate in the 4-inositol position (*Figure 12*) and may induce oligomerization in a manner analogous to the 4-phosphate. In contrast, a phosphate at the 6-inositol position is unlikely to be tolerated without reorganization of the binding site due to a steric clash with S35. Overall, a combination of phosphate groups on the inositol ring in position 5, together with an additional phosphate in position 3 or 4, appears to be capable of oligomerization. Furthermore, PIPs harboring phosphate groups in positions 3, 4, and 5 should be well tolerated in the grip.

Our structure of the $NaD1:PIP_2$ complex suggests that formation of the cationic grip via a NaD1 dimer is important for the formation of a functional $PIP_2$ binding site. Notably, this dimeric arrangement is different to the dimers we previously observed for NaD1 alone (*Lay et al., 2012*). Although one of the two dimer configurations observed was based on an interface formed by two opposing β1 strands, the overall configuration places the β2–β3 loops at opposite ends of the dimer (*Lay et al., 2012*), so that the 'cationic grip' is absent. Consequently, a substantial reorientation is required to convert the NaD1 dimer observed in the absence of $PIP_2$ into a configuration that can engage $PIP_2$ and oligomerize.

Although CAP:lipid interactions (e.g., nisin:lipid II, plectasin:lipid II) have been investigated using structural methods (*Hsu et al., 2004*; *Schneider et al., 2010*), the formation of defensin:ligand oligomers as observed for $NaD1:PIP_2$ is unique. NaD1 engages $PIP_2$ in a cooperative manner where three NaD1 monomers form a complete $PIP_2$ binding site, with each NaD1 monomer participating in the formation of three distinct $PIP_2$ binding sites. This ability to participate in the engagement of multiple $PIP_2$ molecules is mediated by two key basic residues, K4 and R40. The lantibiotic peptide nisin binds specifically to the membrane-bound cell wall precursor lipid II [undecaprenyl-pyrophosphoryl-Mur-NAc-(pentapeptide)-GlcNAc], leading to pore formation and permeabilization of the bacterial cytoplasmic membrane (*Sahl and Bierbaum, 1998*; *Wiedemann et al., 2001*). Structural studies have revealed that nisin forms a 1:1 complex with lipid II (*Hsu et al., 2004*), however the precise contribution of lipid II binding to plasma membrane permeabilization is not fully established. Interestingly, the report on the nisin:lipid II structure indicated that nisin precipitated upon lipid II addition and the structure could only be determined by dissolving the nisin:lipid II complex in DMSO (*Hsu et al., 2004*). It is conceivable that nisin also forms oligomers that were present in the precipitated sample that was discarded. The fungal defensin plectasin is another antimicrobial peptide that requires lipid II binding for its biological activity (*Schneider et al., 2010*). NMR spectroscopy was used to map a putative lipid II binding site on plectasin, however the interaction appears again to be a 1:1 complex, with no

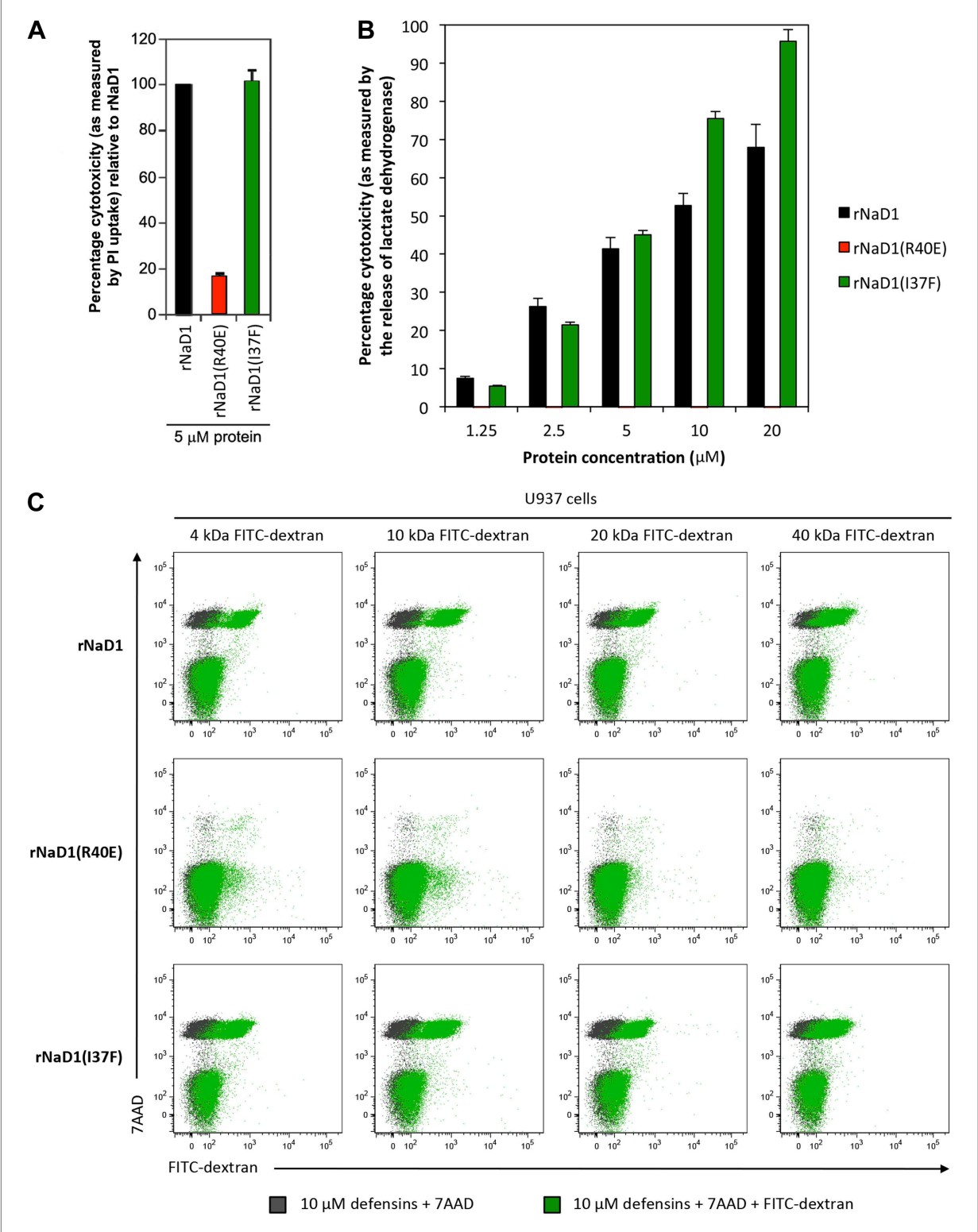

**Figure 11**. Permeabilization of U937 cells is impaired in rNaD1(R40E). Ability of rNaD1, rNaD1(R40E) and rNaD1(I37F) to permeabilize U937 cells as assessed by (**A**) PI uptake, (**B**) LDH release, and (**C**) FITC-dextran binding assays. Error bars in **A** and **B** indicate SEM (n = 3). Data in **A**–**C** are representative of at least two independent experiments.

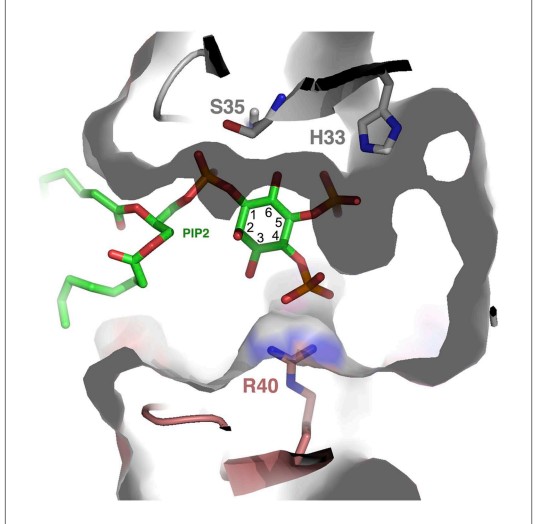

**Figure 12**. PIP$_2$ fit into the NaD1 'cationic grip'. Cut-away of PIP$_2$ bound in the NaD1 cationic grip. NaD1 dimer surface is shown in gray, PIP$_2$ in green and select NaD1 residues are shown as stick representation. NaD1 monomeric chains are colored in gray and salmon. Carbon atoms in the inositol ring are numbered (1–6).

suggestion of oligomer formation. Furthermore, plectasin does not permeabilize target microbial membranes and acts by directly targeting bacterial cell wall biosynthesis (*Schneider et al., 2010*).

Amongst the many described phospholipid-binding proteins, a number of different domain structures have been defined that exhibit stereo-specific recognition of specific phosphoinositide head groups in the context of cellular membrane surfaces. These include the pleckstrin homology (PH), 'Fab1, YOTB, Vac1, EEA1' (FYVE), and Phox-homology (PX) domains (*Bravo et al., 2001*; *Lemmon, 2008*), with all three showing structural similarities to that we describe for PIP$_2$ binding by NaD1. In each of these domains, phosphoinositide binding is mediated through pockets that are strategically lined with basic residues formed by two distal β loops. Pleckstrin homology domains consist of two perpendicular antiparallel β-sheets followed by a C-terminal amphipathic helix, with the canonical phosphoinositide binding pocket comprising basic residues derived from two distal β-loops on the two β-sheets (*Lietzke et al., 2000*). PX domains contain a triple-stranded antiparallel β-sheet followed by a helical subdomain made up of four α-helices, with the β1 and β2 loops together with α-helix 3 forming a positively-charged pocket responsible for binding phosphoinositides (*Bravo et al., 2001*). FYVE domains are small cysteine-rich Zn$^{2+}$ binding domains, consisting of two β-strands followed by a small C-terminal α-helix, with the binding pocket for the inositol head group of PI(3)P comprising basic clusters at either end of the β1 strand, one of which possesses a common (R/K)(R/K)HHCR motif (*Kutateladze and Overduin, 2001*). The topology of the phosphoinositide binding pocket formed by the NaD1 dimer is similar to that of the PH domains, however the formation of the pocket is unique and involves the symmetrical juxtaposition of identical β2–β3 loops (comprised of KILRR) by dimerization of two NaD1 monomers. Thus, despite the very different overall folds of NaD1 and the other phosphoinositide binding domains, key structural features that govern how they recognize phosphoinositides are generally conserved.

It is of interest that a recent report describes the ability of human α-defensin 6 to self-assemble into high-order oligomers termed nanonets (*Chu et al., 2012*). In contrast to NaD1:PIP$_2$ oligomers, these fibril-like structures appear to form without involvement of defined extrinsic ligands and rely on stochastic binding to bacterial surface proteins to initiate self-assembly. Although composed of defensins, these nanonets do not harbor direct antibacterial activity per se, but rather act by trapping bacteria to prevent cellular adhesion and invasion (*Chu et al., 2012*). Together with our findings, these studies indicate that defensins are able to form different fibrils or oligomers for diverse functions in innate immunity.

Certain CAPs and cell penetrating peptides have been reported to penetrate biological membranes with or without membrane permeabilization (*Henriques et al., 2006*; *Ashida et al., 2011*). Although it is yet to be elucidated how NaD1 could pass through the plasma membrane to interact with phospholipids at the inner leaflet, our data provide several significant insights. NaD1 permeabilizes mammalian cells by forming a novel phosphoinositide recognition complex associated with the formation of membrane blebs and membrane rupture, possibly involving disruption of cytoskeleton-membrane interactions through the binding of PIP$_2$ at the inner leaflet (*Figure 13*). This novel mechanism of cell lysis is distinct from that proposed for other CAPs that act via pore formation or a non-specific charge-based interaction with the plasma membrane (*Brogden, 2005*) and the well-defined pore-forming ability of cholesterol-dependent cytolysins (*Rossjohn et al., 1997*), perforin (*Law et al., 2010*) or complement membrane attack complex (*Hadders et al., 2007*). These findings not only reveal a new mechanism of cell lysis but also uncover a potential evolutionarily conserved

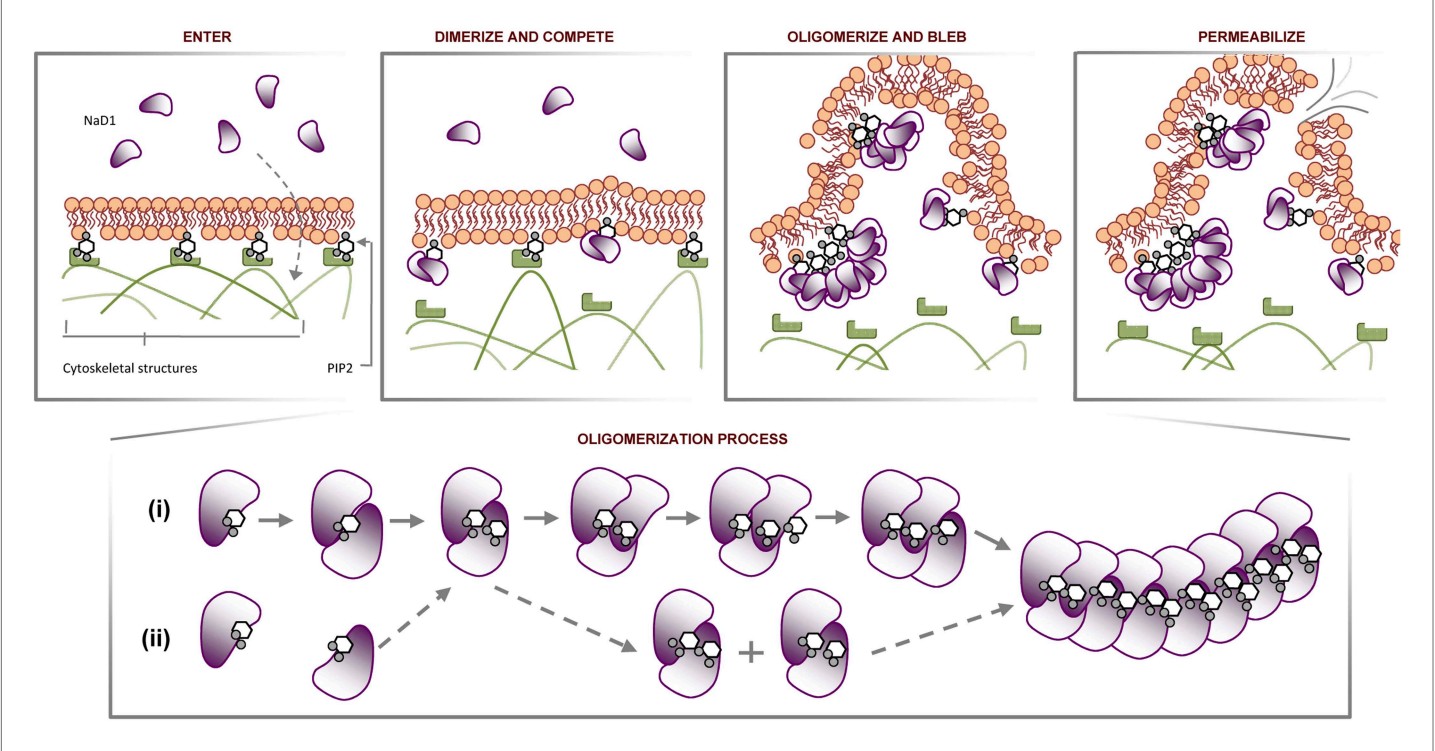

**Figure 13**. Proposed molecular mechanism of NaD1-mediated tumor cell lysis. Schematic representation of NaD1-induced membrane blebbing and permeabilization. The assembly of NaD1:PIP$_2$ oligomer can potentially be formed by (i) sequential recruitment of a NaD1 monomer followed by a PIP$_2$ molecule or (ii) dimerization of two single NaD1:PIP$_2$ complex followed by the recruitment of NaD1:PIP$_2$ dimers.

innate defense mechanism that can target cell membranes through the recognition of a 'phospholipid pattern/code'.

## Materials and methods

### Purification of NaD1

NaD1 was isolated from its natural source, whole *Nicotiana alata* flowers, as described previously (*van der Weerden et al., 2008*).

### Preparation of reduced and alkylated NaD1 (NaD1$_{R\&A}$)

Reduction and alkylation of NaD1 was performed as described previously (*van der Weerden et al., 2008*).

### BODIPY labeling of NaD1

BODIPY labeling of NaD1 was performed as described previously (*van der Weerden et al., 2010*).

### Protein-lipid overlay assay

NaD1 binding to lipids spotted on Membrane Lipid Strips, PIP Strips or SphingoStrips was performed according to manufacturer's instructions (Echelon Biosciences, Salt Lake City, UT). Briefly, lipid strips were incubated with PBS/3% fatty acid-free BSA for 120 min at RT to block non-specific binding. The lipid strips were then incubated with NaD1 or NaD1 mutants (1 µg/ml) diluted in PBS/1% fatty acid-free BSA for 60 min at 4°C and washed thoroughly for 60 min at RT with PBS/0.1% Tween-20. Membrane-bound NaD1 was detected by probing the lipid strips with 2 µg/ml of a protein A purified rabbit anti-NaD1 antibody (*van der Weerden et al., 2008*) diluted in PBS/1% fatty acid-free BSA for 60 min at 4°C, followed by a HRP-conjugated donkey anti-rabbit Ig antibody (GE Healthcare, Buckinghamshire, United Kingdom) diluted to 1:2000 in PBS/1% fatty acid-free BSA for 60 min at 4°C. After each antibody incubation, the lipid strips were washed for 30 min at RT with PBS/0.1% Tween-20.

Chemiluminescence was detected using the enhanced chemiluminescence reagent (GE Healthcare) and developed using Hyperfilm (GE Healthcare). Relative binding of proteins to the lipids was quantitated by densitometry using ImageJ software (National Institutes of Health, Bethesda, MD; http://rsb.info.nih.gov/ij). The data were normalized to the PtdIns(4,5)$P_2$ (for PIP Strips and Membrane Strips) or sulfatide (for SphingoStrips).

## Generation of liposomes and liposome pull-down assays

Liposomes were prepared as described previously (*Zhang et al., 2001*) using lipids purchased from Avanti Polar Lipids (Alabaster, AL); L-α-phosphatidylcholine (PC, chicken egg), L-α-phosphatidyl-DL-glycerol (PG, chicken egg), L-α-phosphatidylethanolamine (PE, chicken egg), L-α-phosphatidylinositol (PI, bovine liver), L-α-phosphatidylserine (PS, porcine brain), and L-α-phosphatidylinositol-4,5 bisphosphate ($PIP_2$, porcine brain). Lipids (dissolved in chloroform) and $PIP_2$ (dissolved in chloroform, methanol and water) were combined with the desired ratio of lipid components (PC:PE:PS:PI 50:30:10:10, PC:PE:PS:PI:$PIP_2$ 50:30:10:8:2, PC:PG 75:25, PC:PG:$PIP_2$ 75:20:5). The lipid mixture was dried under a stream of nitrogen gas followed by further drying under a vacuum for 3 hr. The lipid films were rehydrated in 500 µl of 50 mM HEPES (pH 7.0) to a concentration of 14 mg/ml for 2 hr with occasional vortexing. Lipid mixtures were freeze-thawed five times before sonicating for 8 min until the mixture cleared. Liposomes were washed twice in 50 mM HEPES (pH 7.0) prior to liposome binding assay. 50 µl of 14 mg/ml liposomes were incubated with 0.5 µg of NaD1 for 30 min at 25°C. Sample was pelleted by centrifugation at 16000×*g* and 30 µl of supernatant collected. The pellet was washed twice with 100 µl of 50 mM HEPES pH 7.0. Supernatant and pellet samples were analyzed for the presence of protein by SDS-PAGE and immunoblotting using a rabbit anti-NaD1 antibody as described for the protein-lipid overlay assay.

## Crystallographic methods

The NaD1:$PIP_2$ complexes were generated by mixing NaD1 at 10 mg/ml and $PIP_2$ at a molar ratio of 1:1.2. Crystals were grown in sitting drops at 20°C in 0.2 M ammonium sulfate, 7% PEG 3350, 32% MPD, and 0.1 M imidazole pH 7. Diffraction data were collected from crystals flash cooled in mother liquor at 100 K at the Australian Synchrotron (beamline MX2) and processed with Xds (*Kabsch, 2010*). The structure was solved by molecular replacement with PHASER (*Storoni et al., 2004*) using the structure of NaD1 (*Lay et al., 2012*) as a search model. The final model was built with Coot (*Emsley and Cowtan, 2004*) and refined with Phenix (*Adams et al., 2010*) to a resolution of 1.6 Å. All data collection and refinement statistics are summarized in *Table 1*. Refinement yielded $R_{work}$ and $R_{free}$ values of 15.5% and 18.4%, respectively. All programs were accessed via the SBGrid suite (*Morin et al., 2013*). The coordinates have been deposited in the Protein Data Bank (accession code 4CQK). Figures were prepared using PyMol.

## Cross-linking studies

NaD1 at 1 mg/ml (5 µl) was incubated with 2.3, 0.46, and 0.092 mM $PIP_2$ (5 µl) at room temperature for 30 min. Protein complexes were cross-linked through primary amino groups by the addition of 12.5 mM bis[sulfosuccinimidyl] suberate ($BS^3$; 10 µl) in a buffer containing 20 mM sodium phosphate and 150 mM NaCl, pH 7.1, at room temperature for 30 min. Samples were reduced and denatured, and subjected to SDS-PAGE prior to Coomassie Brilliant Blue staining.

## Transmission electron microscopy (TEM)

TEM imaging was performed according to the procedure described by *Adda et al. (2009)*. In brief, samples (10 µl) were applied to 400-mesh copper grids coated with a thin layer of carbon for 2 min. Excess material was removed by blotting and samples were negatively stained twice with 10 µl of a 2% (wt/vol) uranyl acetate solution (Electron Microscopy Services, Hatfield, PA). The grids were air-dried and viewed using a JEOL JEM-2010 transmission electron microscope operated at 80 kV.

## Expression of rNaD1, rNaD1(R40E), and rNaD1(I37F) in *Pichia pastoris*

Recombinant NaD1 and the point mutants rNaD1(R40E) and rNaD1(I37F) were cloned, expressed, and purified from the methylotropic yeast *Pichia pastoris* as described in *Lay et al. (2012)*.

## Fungal growth inhibition assays

The ability of rNaD1, rNaD1(I37F), and rNaD1(R40E) to inhibit the growth of *F. oxysporum* f. sp. vasinfectum was assessed as described in *van der Weerden et al. (2008)*, except that 6400 spores/well were

used and growth was assessed after 48 hr. Each test was performed in triplicate. For data analysis, Prism 5 software (GraphPad Software Inc., San Diego, CA) was used to plot 4-parameter sigmoidal curves through the data.

## Cell lines

Human epithelial cervical cancer (HeLa) cells, prostate cancer (PC3) cells, and monocytic lymphoma (U937) cells were cultured in RPMI-1640 medium (Invitrogen, Carlsbad, CA). All culture media were supplemented with 5–10% fetal calf serum, 100 U/ml of penicillin, and 100 µg/ml of streptomycin (Invitrogen). Cell lines were cultured at 37°C in a humidified atmosphere containing 5% $CO_2$. Adherent cell lines were detached from the flask by adding a mixture containing 0.25% trypsin and 0.5 µM EDTA (Invitrogen).

## Propidium iodide (PI) uptake assay

Flow cytometry-based PI uptake assay was performed to analyze the ability of NaD1 and related defensins to permeabilize tumor cells. Unless stated otherwise, U937 cells were suspended to $1 \times 10^6$ cells/ml in 0.1% BSA/RPMI-1640 and incubated with protein samples at 37°C for 30 min. Samples were added to PBS containing a final concentration of 1 µg/ml PI (Sigma-Aldrich, St Louis, MO) and placed on ice. Samples were then analyzed immediately using the BD FACSCanto II Flow Cytometer and BD FACSDiva software v6.1.1 (BD Biosciences, St Jose, CA). The resultant flow cytometry data were analyzed using FlowJo software v8.8.6 (Tree Star, Ashland, OR). Cells were gated appropriately based on forward scatter and side scatter and cell permeabilization was defined by PI-positive staining.

## FITC-dextran uptake assay

U937 cells and protein samples were prepared as per the PI uptake assay, with the exception that 100 µg/ml of FITC-dextran (4, 10, 20, or 40 kDa, Sigma-Aldrich) was present during the incubation at 37°C for 30 min. Samples were washed twice with 0.1% BSA/PBS to remove unbound FITC-dextran and added to PBS containing a final concentration of 1 µg/ml 7-aminoactinomycin D (7AAD) prior to analysis using the BD FACSCanto II Flow Cytometer and BD FACSDiva software. The resultant flow cytometry data were analyzed using FlowJo software.

## ATP bioluminescence assay

ATP bioluminescence assay (Roche, Mannheim, Germany) was used according to manufacturer's instructions to measure the release of ATP from permeabilized tumor cells following treatment with defensins. Briefly, U937 cells were suspended to $1 \times 10^6$ cells/ml in 0.1% BSA/PBS and mixed with luciferase reagent at a ratio of 4:5 (vol:vol). The mixture of cells and luciferase reagent were added simultaneously to each well containing protein samples and luciferase activity was measured immediately on a SpectraMax M5e plate reader (Molecular Devices, Sunnyville, CA) at RT for 30 min with readings taken at 30 s intervals. The resultant data were analyzed using SoftMaxPro 5.2 software (Molecular Devices).

## Lactate dehydrogenase (LDH) release assay

LDH cytotoxicity assay kit II (Abcam, Cambridge, United Kingdom) was used according to manufacturer's instructions to detect the release of the cytosolic enzyme, LDH, from U937 cells following treatment with defensins. Briefly, U937 cells were suspended at a cell concentration of $1 \times 10^6$ cells/ml in 0.1% BSA/RPMI-1640 and incubated with protein samples at 37°C for 30 min. Cells were then pelleted by centrifugation at $600 \times g$ and the supernatant was added to LDH reaction mix for 30 min at RT. The absorbance of the enzymatic product at 450 nm was measured using a SpectraMax M5e plate reader, with the resultant data analyzed using SoftMaxPro 5.2 software.

## MTT cell viability assay

Mammalian cells were seeded in quadruplicate into wells of a flat-bottomed 96-well microtitre plate (50 µl) at various densities starting at $2 \times 10^6$ cells/ml. Four wells containing complete culture medium alone were included in each assay as a background control. The microtitre plate was incubated overnight at 37°C under a humidified atmosphere containing 5% $CO_2$/95% air, prior to the addition of complete culture medium (100 µl) to each well and further incubated at 37°C for 48 hr. Optimum cell densities (30–50% confluency) for cell viability assays were determined for each cell line by light microscopy. Mammalian cells were seeded in a 96-well microtitre plate (50 µl/well) at an optimum

density determined in the cell optimization assay as above. Background control wells (n = 8) containing the same volume of complete culture medium were included in the assay. The microtitre plate was incubated overnight at 37°C, prior to the addition of NaD1 at various concentrations and the plate was incubated for a further 48 hr. The cell viability 3-(4, 5-dimethyl-2-thiazolyl)-2, 5-diphenyl-2H-tetrazolium bromide (MTT, Sigma-Aldrich) assay was performed as follows: the MTT solution (1 mg/ml) was added to each well (100 μl) and the plate incubated for 2–3 hr at 37°C under a humidified atmosphere containing 5% $CO_2$/95% air. Subsequently, the media was removed and replaced with dimethyl sulfoxide (100 μl, DMSO, Sigma-Aldrich) and placed on a shaker for 5 min to dissolve the tetrazolium salts. Absorbance of each well was measured at 570 nm and the $IC_{50}$ values (the protein concentration to inhibit 50% of cell growth) were determined using OriginPro software v8.1.13.88 (OriginLab Corporation, Northampton, MA).

## Confocal laser scanning microscopy (CLSM)

Live imaging was performed on a Zeiss LSM 510 or LSM 780 confocal microscope using a 40× or 63× oil immersion objective in a 37°C/5% $CO_2$ atmosphere. Adherent cells were cultured on coverslips prior to imaging, while non-adherent cells were immobilized onto 10% poly-L-lysine-coated coverslips. All cell types were prepared for imaging in RPMI medium containing 0.1% BSA and 1–2 μg/ml PI. NaD1, BODIPY-NaD1, and FITC-Dextran (100 μg/ml) was added directly to the imaging chamber via a capillary tube. In certain experiments, cells were either stained with PKH67 (Sigma-Aldrich) or transfected with a plasmid construct for free GFP or GFP-PH(PLCδ) using Lipofectamine 2000 Reagent (Invitrogen) as per manufacturer's instructions prior to imaging. The images were analyzed using ImageJ software or Zen software (Zeiss, Oberkochen, Germany). For quantification of CLSM experiments involving transfected cells, the effects of NaD1 on HeLa cells were observed over a 15 min timeframe from the time of NaD1 addition. Non-expressing cells were excluded from analysis.

## Acknowledgements

We thank members of the Hulett and Kvansakul laboratories for their suggestions. We also thank S Cole for assistance with the purification of native NaD1; A Bojceski for assistance with the protein-lipid overlay assays; M Wong for assistance with flow cytometry; J Dyson and C Mitchell for suggestions and providing the construct for GFP-PH(PLCδ); T Brown for assistance with CLSM; the Bio21 Collaborative Crystallisation Centre for assistance with crystallization; MX2 staff at the Australian Synchrotron for assistance with X-ray diffraction data collection; SBGrid consortium for software support. This work was supported by Balmoral Australia Pty Ltd and Hexima Ltd.

## Additional information

### Competing interests

NLW: Head of Technology Commercialisation Hexima Ltd. MAA: Chief Science Officer Hexima Ltd. MDH: Vice President of Research Hexima Ltd. The other authors declare that no competing interests exist.

### Funding

| Funder | Grant reference number | Author |
| --- | --- | --- |
| National Health and Medical Research Council Career Development Fellowship | 637372 | Marc Kvansakul |
| National Health and Medical Research Council Project Grant | APP1007918 | Marc Kvansakul |
| Australian Research Council Discovery Project Grant | DP120102694 | Nicole L van der Weerden, Marilyn A Anderson, Mark D Hulett |
| Australian Research Council Fellowship | FT130101349 | Marc Kvansakul |

The funders had no role in study design, data collection and interpretation, or the decision to submit the work for publication.

## Author contributions

IKHP, AAB, FTL, MK, MDH, Conception and design, Acquisition of data, Analysis and interpretation of data, Drafting or revising the article; GDM, CGA, JAEP, TKP, GFR, JAW, PKV, Acquisition of data, Analysis and interpretation of data; NLW, MAA, Acquisition of data, Analysis and interpretation of data, Drafting or revising the article, Contributed unpublished essential data or reagents

## Additional files

### Major dataset

The following dataset was generated:

| Author(s) | Year | Dataset title | Dataset ID and/or URL | Database, license, and accessibility information |
|---|---|---|---|---|
| Lay FT, Mills GM, Poon IKH, Baxter AA, Hulett MD, Kvansakul M | 2014 | Crystal structure of ligand-bound NaD1 | 4CQK; http://www.rcsb.org/pdb/search/structidSearch.do?structureId=4CQK | Publicly available at the RCSB Protein Data Bank (http://www.rcsb.org/). |

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
