## [Decision Letter]

Thank you for sending your work entitled “PtdIns(4,5)P_2_-mediated oligomerization of the defensin NaD1 represents a novel mechanism of cell lysis” for consideration at *eLife*. Your article has been favorably evaluated by a Senior editor, John Kuriyan, and 2 reviewers.

The editor and the other reviewers discussed their comments before we reached this decision, and the editor has assembled the following comments to help you prepare a revised submission.

The manuscript by Poon et al. describes a thorough structure-function study of the ornamental tobacco defensin NaD1. Defensins belong to the group of cationic antimicrobial peptides, which function as ubiquitously expressed innate immune molecules that often destabilize membrane structure as the detrimental step for the target cells. While plant defensins share a common structure, their primary sequences are less conserved, which is likely linked to their apparent ligand specificity. In previous work, this research team determined the basic structure and cellular mode of action of NaD1. Here, they describe several new discoveries that present a large step forward in our molecular understanding of these peptides. In particular, this is the first report of any defensin making a specific interaction with a phosphoinositide, and the structural information provided by the work is really first-rate and informative.

The authors identify the membrane lipid PIP_2_ as the high-affinity ligand for NaD1 in fungal and mammalian cells. Furthermore, they demonstrate, based on a novel crystal structure and biochemical data, that NaD1 oligomerizes upon PIP_2_ binding and this step significantly increases NaD1's cytolytic activity. While the PIP_2_ binding sites created by NaD1 oligomers has features seen in other PIP_2_ binding modules, NaD1 adopts a fold distinct from the other well-established protein domains. Finally, elegant experiments establish the timing of events triggered by NaD1, in particular membrane blebbing, cell permeabilization and cytolysis. Taken together, this is a well-written and executed study that makes a significant contribution to the field. Given the broad expression of defensins and the general importance of PIP2 for cellular physiology, this work should be of great interest to the diverse readership of *eLife*.

The reviewers have identified one issue that should perhaps be addressed experimentally before submitting a revised paper, and hopefully this should cause little difficulty. The revised manuscript will be considered by the editor without further input from the reviewers, and so a clear description of what is done in the revision would help make a quick decision.

Important experimental issue:

The discussion on PIP specificity relies heavily on the protein-overlay assays that utilize PIP Strips, which are notoriously unreliable. The manner in which the lipids are displayed on these blots bears little resemblance to how they are displayed in lipid bilayers. The authors have presented single blots and their quantitation is simply densitometry of these single strips. The authors should give the readers a realistic indication as to how reproducible these blots are. There should be errors indicated on the quantitation that refer to biological replicates. These strips can vary wildly from one experiment to another and from batch to batch (they were commercially supplied). Ideally, it would be helpful if the authors used measurements on liposomes, but this is not essential. But, unless validated by liposome binding experiments, the authors should refrain from quantitative arguments. Along these lines, quantification of NaD1 binding (wild type vs mutant) to PIP-containing liposomes is also desirable since it may reveal significant differences that are not apparent in the more artificial PIP Strip assays.

---

## [Author Response]

*The discussion on PIP specificity relies heavily on the protein-overlay assays that utilize PIP Strips, which are notoriously unreliable. The manner in which the lipids are displayed on these blots bears little resemblance to how they are displayed in lipid bilayers. The authors have presented single blots and their quantitation is simply densitometry of these single strips. The authors should give the readers a realistic indication as to how reproducible these blots are. There should be errors indicated on the quantitation that refer to biological replicates. These strips can vary wildly from one experiment to another and from batch to batch (they were commercially supplied). Ideally, it would be helpful if the authors used measurements on liposomes, but this is not essential. But, unless validated by liposome binding experiments, the authors should refrain from quantitative arguments. Along these lines, quantification of NaD1 binding (wild type vs mutant) to PIP-containing liposomes is also desirable since it may reveal significant differences that are not apparent in the more artificial PIP Strip assays*.

The reviewers raise an excellent point for the protein overlay lipid strips in that we should provide an indication of reproducibility by including errors on the quantitation referring to biological replicates. We have performed statistical analysis on biological replicates for the relevant experiments, and have amended Figure 1—figure supplement 1 and Figure 6—figure supplement 1, accordingly.

The reviewers also suggested that it would be helpful (but not essential) to use liposome-binding experiments to quantify NaD1 binding (wild-type vs mutant). We agree that these data would provide support to the lipid-blot data and have attempted the experiments. However, we were not able to adequately resolve a number of technical challenges in a reasonable time frame, which was exacerbated by the subsequent need for the production of additional recombinant proteins. The reviewers suggested that if were unable to provide quantitative liposome pull-down data, we should refrain from quantitative arguments. As such, we have modified the text accordingly by replacing several statements, as follows:

A) In the first Results paragraph “Noticeably, NaD1 bound strongest to PIP2 relative to all other lipids tested....”, has been changed to “Interestingly, NaD1 bound the functionally important phospholipid PIP2....”

B) In the Results section entitled “PIP2 binding and oligomerization of NaD1 are critical for fungal cell killing”, “...resulted in a dramatic reduction in the binding of PI(4)P” has been replaced with “However, it did result in reduced binding of PI(4)P...”